# A paralog-specific role of COPI vesicles in the neuronal differentiation of mouse pluripotent cells

Manu Jain Goyal[1,2], Xiyan Zhao[1,2], Mariya Bozhinova[1,2], Karla Andrade-López[1,2], Cecilia de Heus[3], Sandra Schulze-Dramac[4], Michaela Müller-McNicoll[4] ⓘ, Judith Klumperman[3], Julien Béthune[1,2] ⓘ

Coat protein complex I (COPI)–coated vesicles mediate membrane trafficking between Golgi cisternae as well as retrieval of proteins from the Golgi to the endoplasmic reticulum. There are several flavors of the COPI coat defined by paralogous subunits of the protein complex coatomer. However, whether paralogous COPI proteins have specific functions is currently unknown. Here, we show that the paralogous coatomer subunits γ1-COP and γ2-COP are differentially expressed during the neuronal differentiation of mouse pluripotent cells. Moreover, through a combination of genome editing experiments, we demonstrate that whereas γ-COP paralogs are largely functionally redundant, γ1-COP specifically promotes neurite outgrowth. Our work stresses a role of the COPI pathway in neuronal polarization and provides evidence for distinct functions for coatomer paralogous subunits in this process.

## Introduction

In eukaryotes, membrane trafficking of cargo proteins and lipids is crucial to maintain cellular homeostasis and intracellular organelle identity. In the early secretory pathway in mammals, coat protein complex II (COPII) vesicles mediate the export of cargo from the ER to the Golgi apparatus, whereas COPI vesicles promote the retrieval of proteins from the Golgi to the ER and intra-Golgi transport. COPI vesicles are formed at the Golgi through the GTP-dependent recruitment of a coat protein complex, termed coatomer, by the small GTPase ARF1. Once recruited to membranes, coatomer polymerizes to form a lattice that shapes a nascent bud that eventually pinches off as a small-coated vesicle (Béthune & Wieland, 2018). Through sorting signals exposed on their cytoplasmic domains, transmembrane cargo proteins interact with coatomer and are taken up in COPI vesicles (Barlowe & Helenius, 2016). Coatomer is made of seven equimolar COP subunits (α-,β-,β′-,γ-,δ-,ε-, and ζ-COP) that

are highly conserved from yeast to human. In mammals, two coatomer subunits come as two paralogs: γ1-COP and γ2-COP that share 80% protein sequence identity and that are encoded by the *Copg1* and *Copg2* genes (Blagitko et al, 1999), and ζ1-COP and ζ2-COP encoded by the *Copz1* and *Copz2* genes (Futatsumori et al, 2000). In the COPII system, paralogs of the SEC24 subunit expand the cargo repertoire of COPII vesicles by providing distinct binding sites for specific sorting motifs (Mancias & Goldberg, 2007, 2008; Bonnon et al, 2010; Adolf et al, 2016, 2019). By contrast, proteomics profiling of paralog-specific COPI vesicles generated in vitro from HeLa cells revealed no major difference in their cargo content (Adolf et al, 2019) and to date, no specialized function has been ascribed to the paralogous COP subunits, which until now have thus been considered as functionally redundant.

Whereas the general mechanisms of cargo sorting and formation of COPI vesicles are well described, cell type–specific functions are much less well studied. Several lines of evidence, however, suggest tissue-specific functions of the otherwise essential COPI pathway. Indeed, mutations affecting COP subunits have been associated to diseases, notably neurodegenerative disorders (Xu et al, 2010; Izumi et al, 2016; Bettayeb et al, 2016a, 2016b). Moreover, binding of α-COP to the survival motor neuron protein seems to promote neurite outgrowth in motor neurons (Li et al, 2015). As defects in the COPI pathway lead to specific effects in the nervous system, we decided to analyze the expression profile and function of the γ-COP paralogs during neurogenesis. By examining publicly available mRNA expression profiling data, we found that *Copg1*, but not *Copg2*, is up-regulated as mouse embryonic stem cells (mES) differentiate into neurons. We observed the same in murine pluripotent P19 cells, another model cell line for neuronal differentiation. Specific KO of *Copg1* or *Copg2* in P19 cells revealed that whereas both gene disruptions led to slower cell growth, neither of the two paralogs alone is essential for cell viability. Remarkably, whereas KO of *Copg2* did not affect retinoic acid (RA)–mediated P19 cell neuronal differentiation, disruption of *Copg1* led to the formation of loose embryoid bodies (EBs) and to reduced neurite outgrowth. Overexpression of γ2-COP or knock-in (KI) of *Copg2* in the *Copg1*

[1]Junior Research Group, Cluster of Excellence CellNetworks, Heidelberg, Germany [2]Heidelberg University Biochemistry Center, Heidelberg, Germany [3]Section Cell Biology, Center for Molecular Medicine, University Medical Center Utrecht, Utrecht University, Utrecht, The Netherlands [4]RNA Regulation Group, Cluster of Excellence "Macromolecular Complexes," Institute of Cell Biology and Neuroscience, Goethe University Frankfurt, Frankfurt/Main, Germany

Correspondence: julien.bethune@bzh.uni-heidelberg.de

locus revealed that whereas higher expression of γ2-COP can compensate for the loss of γ1-COP for the formation of EBs, γ1-COP is required to promote neurite outgrowth. Altogether, our findings support an essential role of COPI vesicles during neurogenesis and reveal for the first time a paralog-specific function for a COPI protein subunit.

# Results

### Differential expression of *Copg1* and *Copg2* during neuronal differentiation

To investigate a potential paralog-specific function of COP subunits during neurogenesis, we first examined publicly available RNAseq expression profiling data performed on mES, their derived neuronal progenitors and terminal neurons (Tippmann et al, 2012). In the three differentiation stages, *Copg2* was marginally expressed compared with the other COP subunits with expression levels 10–40 times lower than those of *Copz1* (Fig S1). Interestingly, whereas all other COP subunit-coding genes were expressed at similar levels at all three differentiation stages, *Copg1* was strongly up-regulated in terminal neuron (Fig S1) suggesting γ1-COP might exert a unique function during the biogenesis of neurons. To examine whether a differential expression of *Copg1* and *Copg2* is also observed in

another neuronal differentiation system we then analyzed mRNA and protein levels of the two γ-COP paralogs in P19 embryonal carcinoma cells. Compared with mES, P19 cells are closer to epiblast-derived stem cells and are a well-established and robust model system for neuronal differentiation (Morassutti et al, 1994; Staines et al, 1994; Kelly & Gatie, 2017). Similar to mES, P19 cells can be differentiated into neurons applying a two-step differentiation protocol (Fig 1A). Cells are first incubated with RA-containing medium under non-adherent conditions to form EBs (day 2–4). EBs are then dissociated and the cells seeded under adherent conditions to form post-mitotic neurons (day 6–8). After this procedure, cell lysates were collected at different stages of differentiation and analyzed by western blot. As expected, the pluripotency marker Oct-4 was only detectable at day 0, whereas the neuron-specific class III beta-tubulin (βIII-tubulin or Tuj-1 antigen) is strongly up-regulated as the cells progress to day 8 (Fig 1B). The expression levels of different COP subunits were also assessed by western blot (Fig 1B) using previously characterized specific antibodies against each of the γ-COP paralog (Moelleken et al, 2007) and by RT-qPCR (Fig 1C). In the western blot analysis, we surveyed α-, γ-, and δ-COP as they belong to each of the three building blocks that make the stable heptameric complex. Indeed, under high salt conditions (Lowe & Kreis, 1995) or upon chemical modification of exposed lysines (Pavel et al, 1998), coatomer can be separated into the subcomplexes α-/β'-/ε-COP, γ-/ζ-COP, and β-/δ-COP. To

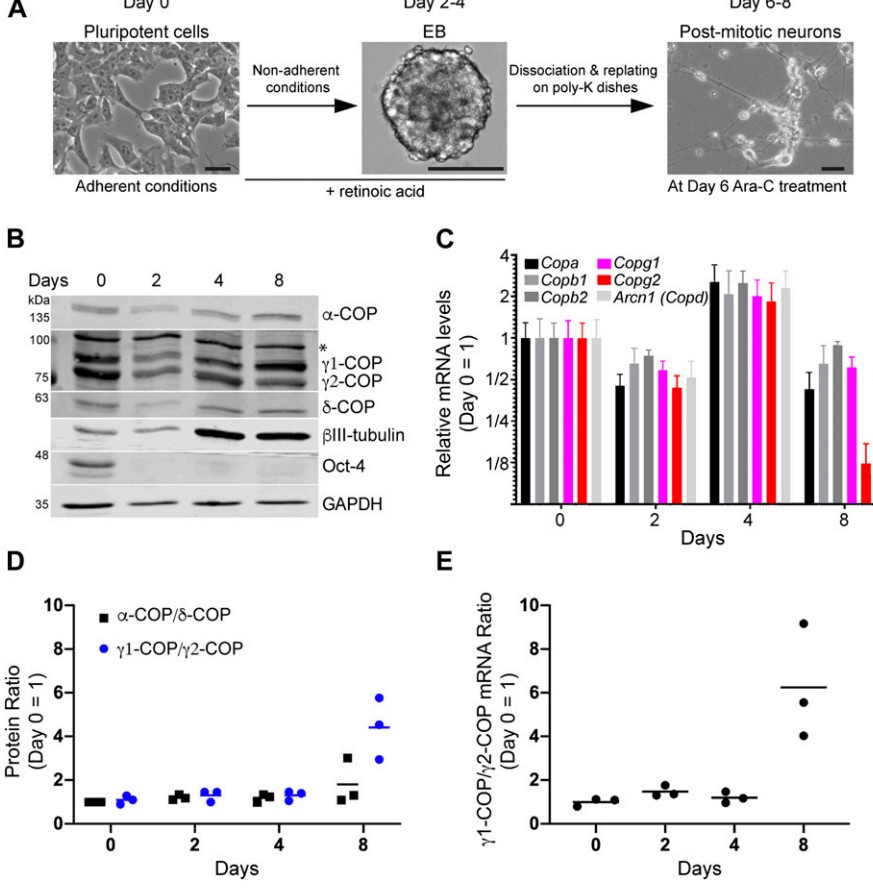

**Figure 1. Differential expression of γ-COP paralogs during neuronal differentiation of P19 cells.**
**(A)** Schematic representation of the two-step differentiation protocol. Scale bar is 50 μm. **(B)** Western blot analysis of P19 cell lysates at different stages of differentiation. The asterisk (*) marks a nonspecific signal. **(C)** Relative expression of mRNAs coding for the indicated COP subunits during neuronal differentiation by RT-qPCR. **(B, C, D, E)** Quantification (n = 3) of the γ1-COP/γ2-COP and α-COP/δ-COP protein and of the γ1-COP/γ2-COP mRNA ratios from (B, C) respectively; bars are means.
Source data are available for this figure.

analyze the relative expression of the two γ-COP paralogs independent of antibody-dependent signal intensities, we considered the ratio of the γ1-COP over γ2-COP signals during differentiation. This ratio was similar in pluripotent cells and EBs; by contrast, there was relatively more γ1-COP in neurons (Fig 1D). This is in contrast to the ratio of α-COP over δ-COP that remained stable over the whole differentiation. This was corroborated at the mRNA levels as *Copg2* mRNA levels decreased at day 8 of differentiation compared with the transcripts of all the other COP subunits, including *Copg1* (Fig 1C and E). Altogether, the expression of γ1-COP is increased compared with γ2-COP during the late stage of neuronal differentiation of pluripotent cells.

## Disruption of *Copg1* or *Copg2* in P19 cells

To study potential paralog-specific functions of COPI vesicles, we then generated *Copg1* and *Copg2* KO P19 cells. To do so, we followed a Cas9-mediated genome editing approach using individual specific single guide RNAs (sgRNAs) directed against either *Copg1* or *Copg2* (see the Materials and Methods section). After clonal selection, we obtained cell lines that showed absence of γ1-COP (*Copg1*$^{-/-}$ cells) or γ2-COP (*Copg2*$^{-/-}$ cells) by western blot analysis (Fig 2A). Moreover, bi-allelic disruption of *Copg1* or *Copg2* was

confirmed by sequencing analysis of the respective genomic loci (Fig S2). Importantly, the KO and clonal selection procedure did not lead to loss of expression of the pluripotency markers Oct-4 and NANOG (Fig 2A), indicating that the KO cell lines are still undifferentiated cells.

## Characterization of *Copg1* or *Copg2* KO in pluripotent P19 cells

As a functional COPI pathway is essential for viability, obtaining individual *Copg1* and *Copg2* KO cells demonstrates that both γ-COP paralogs can fulfill essential functions of COPI vesicles and are thus at least partially redundant. It is, hence, expected that depletion of one or the other γ-COP paralog does not strongly affect the global expression levels of COP subunits and the assembly of the coatomer complex. To test these two predictions, we first analyzed the expression of several COP subunits in WT, and *Copg1* and *Copg2* KO cells. Western blot analysis revealed that expression of α-, δ-, and ε-COP is similar in the three cell lines. As coatomer is an equimolar heptameric complex, the latter observation implies that in the KO cells, the remaining γ-COP paralog is up-regulated. Accordingly, in western blot analyses, γ1-COP expression was reproducibly higher in *Copg2*$^{-/-}$ cells than in the WT situation, and conversely γ2-COP was up-regulated in *Copg1*$^{-/-}$ cells compared with WT (Fig 2B–D).

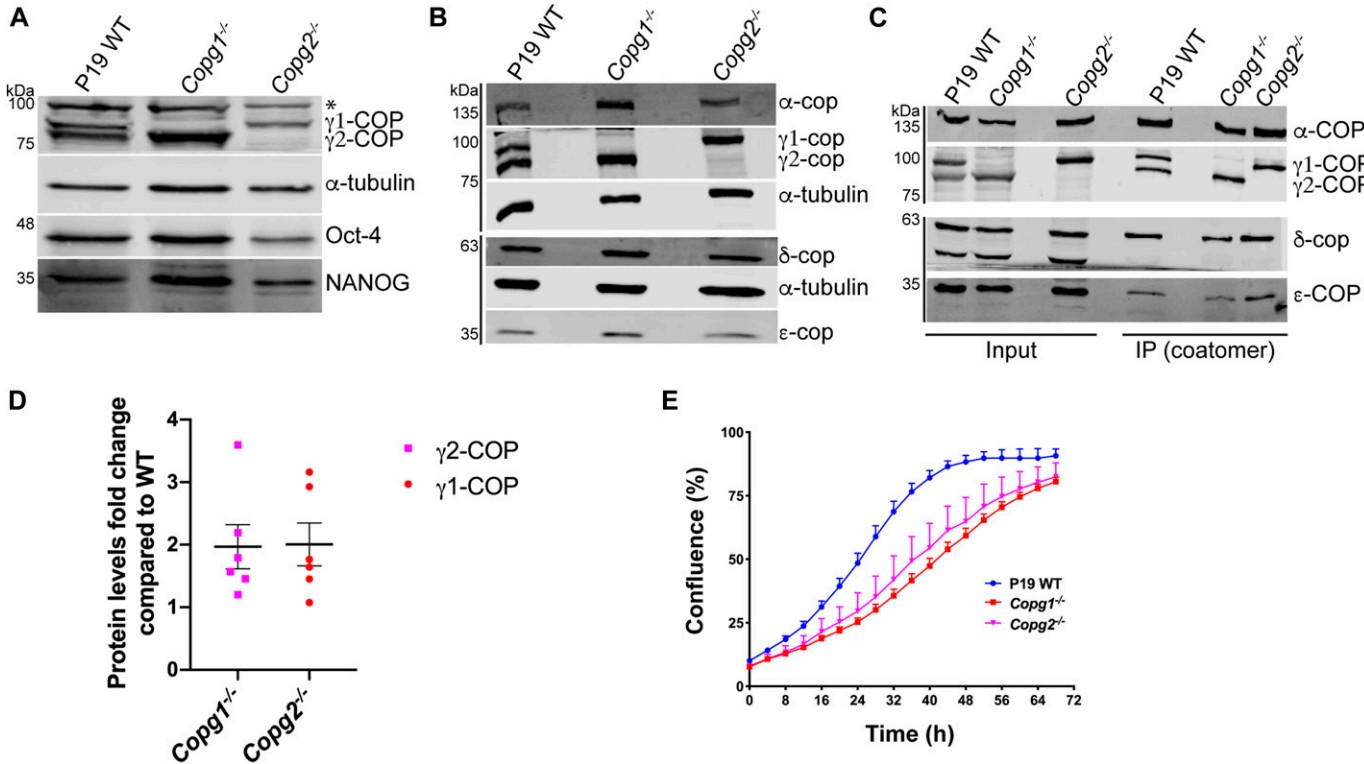

**Figure 2. Characterization of P19 KO cell lines.**
**(A)** Western blot analysis of the expression levels of γ1-COP and γ2-COP, the pluripotency markers Oct-4 and Nanog, and α-tubulin as a loading control in pluripotent P19 WT and KO cell lysates. The asterisk (*) marks a nonspecific signal. **(B)** Western blot analysis of various COP subunits in pluripotent P19 WT and KO cell lysates. **(C)** Immunoprecipitation of native coatomer from P19 WT or KO cell lysates using the CM1 monoclonal antibody. Various COP subunits were analyzed by western blotting as indicated. **(D)** Ratios of γ2-COP western blot signals (normalized to α-tubulin) in *Copg1*$^{-/-}$ over WT cells, and of γ1-COP in *Copg2*$^{-/-}$ over WT cells (bars are means, error bars are SEM, n = 6). **(E)** Growth curves obtained from a real-time proliferation assay in which the occupied area (% confluence) by P19 WT and KO cells was monitored over 72 h. Curves were generated with the IncuCyte software (n = 5, error bars are SEM).
Source data are available for this figure.

Hence, overexpression of one paralog compensates at least partially for the absence of the second one and thus ensures that global coatomer levels are relatively maintained. To verify whether coatomer subunits are correctly incorporated into the complex in the absence of one of the γ-COP paralogs, immunoprecipitation experiments from WT and KO cell lysates were performed using the CM1A10 (CM1) antibody, which specifically recognizes native assembled coatomer (Palmer et al, 1993) probably through its β'-COP subunit (Lowe & Kreis, 1995). As α-, δ-, and ε-COP were precipitated from all three cell lines, this indicates a correct assembly of coatomer in the absence of one specific γ-COP paralog (Fig 2C). Finally, we performed a live cell proliferation assay by measuring the percentage surface occupancy of WT and KO cells in real time as they grow on a cell culture dish under standard conditions. In this assay, both *Copg1* and *Copg2* KO cells showed significantly slower proliferation rates than WT cells (Fig 2E), suggesting that even though both γ-COP paralogs are individually sufficient to mediate essential COPI functions, paralog-specific nonessential functions exist.

### Aberrant Golgi morphology in *Copg1* and *Copg2* KO cells

As COPI vesicles are generated at the Golgi, we then analyzed the morphology of that organelle by EM in WT, *Copg1⁻/⁻*, and *Copg2⁻/⁻* cells (Fig 3A). Quantitative assessment of 30 cell profiles per cell line containing at least one Golgi stack revealed an average higher

number of Golgi stacks per cell in the absence of either γ1-COP or γ2-COP (2.7 Golgi/cell versus 1.8 in WT, Fig 3B). In addition, the average area per Golgi stack was smaller in both KO cell lines ($455–466 \times 10^3$ versus $730 \times 10^3$ $nm^2$ in WT, Fig 3C). Similarly, the average length per cisternae was shorter in the KO cell lines than in the WT cells (167–188 versus 620 nm in WT, Fig 3D). Specific to the depletion of γ1-COP was the observation of an increased number of Golgi peripheral vesicles (27.5% of observed Golgi for which vesicles accounted for more than 50% of the Golgi area versus 13% and 12% WT and *Copg2⁻/⁻* cells, respectively, Fig 3E) indicative of increased Golgi fragmentation. Altogether, both γ1-COP and γ2-COP are necessary for correct assembly and/or maintenance of the Golgi structure.

### Depletion of γ1-COP or γ2-COP does not lead to elevated ER stress

ER stress is known to induce Golgi fragmentation (Nakagomi et al, 2008) and mutations within COP subunits may lead to ER stress (Watkin et al, 2015; Izumi et al, 2016). As ER stress was reported to affect neuronal differentiation of P19 cells (Kawada et al, 2014), we investigated if depletion of γ1-COP or γ2-COP induces ER stress by assessing expression levels of the ER chaperone GRP-78 (alternatively known as Bip) and the transcription factor CHOP, two commonly used ER stress markers (Kennedy et al, 2015). After western blot analysis, we observed similar low expression of

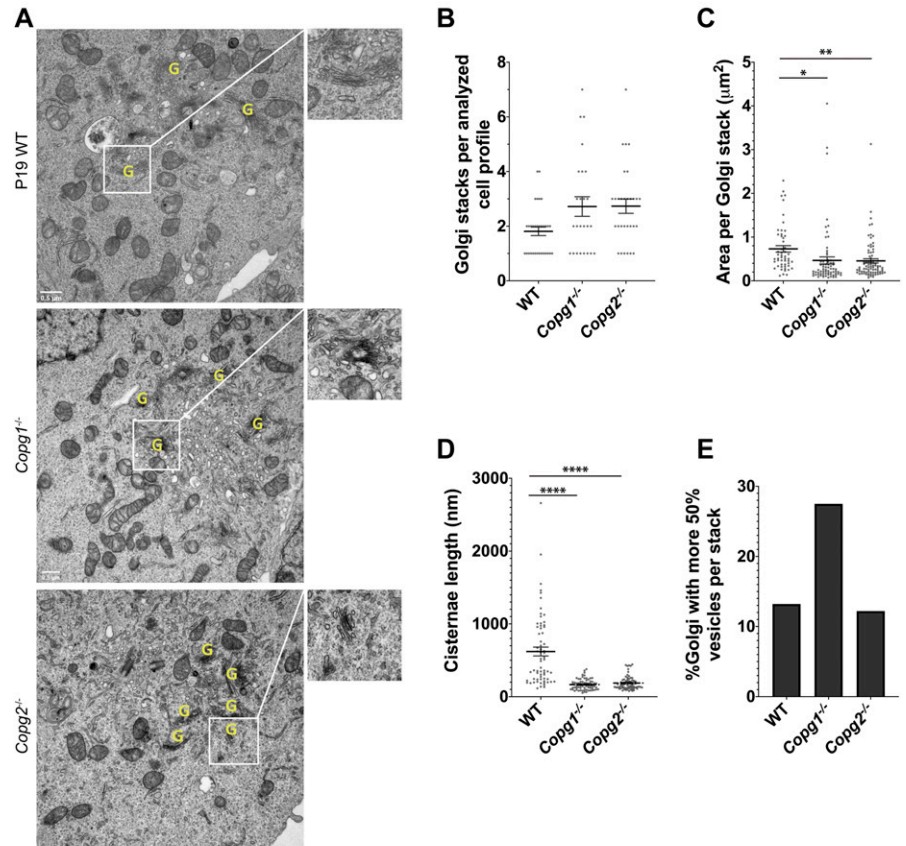

**Figure 3. Depletion of γ1-COP or γ2-COP affects Golgi morphology.**
**(A)** Electron micrographs of thin sections obtained from P19 WT, *Copg1⁻/⁻*, and *Copg2⁻/⁻* cells as indicated. Identified Golgi stacks are indicated by a yellow G. **(B, C, D, E)** Quantification of (B) average number of Golgi stacks per cell, (C) average area per Golgi stack, (D) average cisternae length, and (E) percentage of Golgi stacks with more than 50% of vesicles per stack (*P*-value: * < 0.05, ** < 0.01, **** < 0.0001, n = 30, except for (D): n = 66–72). In (B, C, D) bars are means and error bars are SEM.

GRP-78 and CHOP in WT, *Copg1⁻/⁻*, and *Copg2⁻/⁻* cells (Fig 4A and B). By contrast, induction of GRP-78 and CHOP was clearly visible in cells in which ER stress was induced by tunicamycin, a glycosylation inhibitor that induces the unfolded protein response (Oslowski & Urano, 2011). We note that *Copg1⁻/⁻* cells responded differently to tunicamycin treatment with a lower GRP-78 and a higher CHOP expression when compared with WT and *Copg2⁻/⁻* cells. GRP-78 is a luminal ER protein with a KDEL (Lys-Asp-Glu-Leu) retention signal at its C terminus. The KDEL sequence of escaped ER-resident proteins is recognized at the Golgi by the KDEL receptor which, when complexed with its ligands, is transported back to the ER by COPI vesicles (Jin et al, 2017). CHOP induces the expression of apoptotic genes and is normally up-regulated upon chronic ER stress when the cells cannot cope anymore with the load of misfolded protein in the ER (Oyadomari & Mori, 2004). One possibility that may explain the phenotype of *Copg1⁻/⁻* cells is that in these cells, retrograde transport to the ER might be more affected than in *Copg2⁻/⁻* cells, leading to impaired retention of GRP-78 in the ER. This might render *Copg1⁻/⁻* cells more sensitive to ER stress and thereby lead to a stronger induction of apoptosis via CHOP. At that point, we have not further

explored this possibility. In any case, our data suggest that depletion of γ1-COP or γ2-COP per se does not lead to elevated ER stress.

### Depletion of γ1-COP or γ2-COP does not lead to increased apoptosis

Golgi fragmentation can also be a consequence of apoptosis (Hicks & Machamer, 2005). Because depletion of γ1-COP or γ2-COP led to reduced proliferation rates (Fig 2E), we asked if both proteins are needed to maintain the survival of P19 cells. To do so, we monitored the activity of caspase-3/-7 using FLICA (Fluorescent Labeled Inhibitors of Caspases) probes (Bedner et al, 2000). To distinguish apoptosis from necrosis, cells were counter-stained with the membrane impermeant DNA dye propidium iodide (PI). Thus, flow cytometry analysis allows distinguishing living (FLICA and PI negative), necrotic (FLICA negative and PI positive), early apoptotic cells (FLICA positive and PI negative), and late apoptotic cells (FLICA and PI positive). By contrast to cells treated with staurosporine, an apoptosis-inducing drug through the activation of caspase-3 (Chae et al, 2000), we did not observe elevated apoptotic activity when

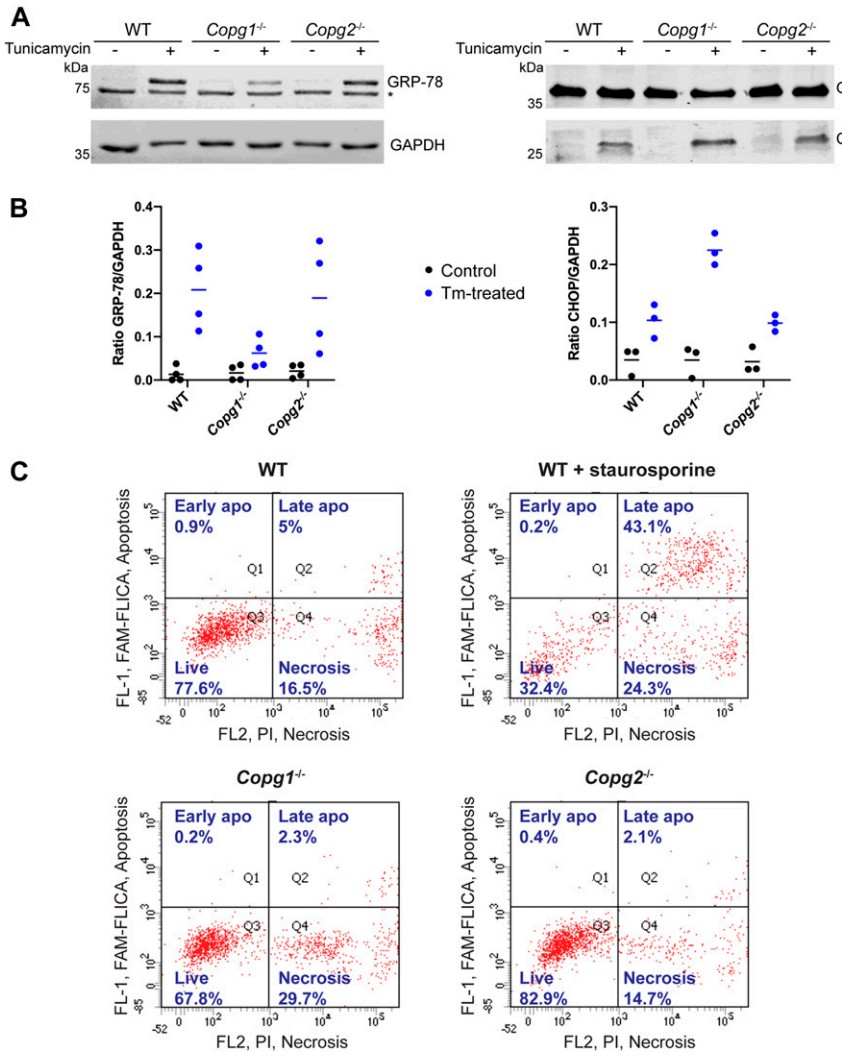

**Figure 4. Depletion of γ1-COP or γ2-COP neither induces ER stress nor apoptosis.**
**(A)** Western blot analysis of the expression of the ER stress markers GRP-78 and CHOP and of GAPDH in P19 WT, *Copg1⁻/⁻*, and *Copg2⁻/⁻* cells. When indicated, ER stress was induced with tunicamycin. The asterisk indicates a nonspecific signal. **(A, B)** Quantification of (A). Tm, tunicamycin. **(C)** Dual fluorescence flow cytometry assay to detect activated caspase-3/7 (FL1, FAM-FLICA) and membrane damage (FL2, propidium iodide) in P19 WT, *Copg1⁻/⁻*, and *Copg2⁻/⁻* cells. Apoptosis (apo) was induced in WT cells with staurosporine as indicated. The percentage of cells detected in each quadrant is indicated (n = 4,686–7,145 cells).
Source data are available for this figure.

comparing *Copg1*$^{-/-}$ and *Copg2*$^{-/-}$ to WT cells (Fig 4C). Hence, depletion of γ1-COP or γ2-COP does not lead to increased apoptosis.

## Expression of γ1-COP is necessary for the formation of tight embryonic bodies

We went on to analyze whether γ-COP paralogs have specific functions during neurogenesis. To address this, P19 WT and KO cells were submitted to the two-step differentiation protocol described in Fig 1A, in which cells are first led to aggregate under non-adherent conditions to form EBs. In this first step, cell aggregation in EBs is essential for the subsequent formation of RA-induced P19 neurons (Jones-Villeneuve et al, 1982; Pachernik et al, 2005). In initial experiments, we performed EB formation in bacterial dishes as described in the original differentiation protocol (Lee et al, 2007). Under these conditions, we noticed that EBs generated from *Copg1*$^{-/-}$ cells seemed smaller and exhibited a less compact morphology than EBs from WT and *Copg2*$^{-/-}$ cells. However, as EBs were free floating in the dishes, it was difficult to obtain pictures and to perform quantitative analyses. To investigate EB morphology in a more controlled way, we turned to a hanging drop protocol in which 200 dissociated cells are seeded in hanging drops of 20 µl RA-containing cell culture medium. The drops were then analyzed by light microscopy at day 2 and day 4 of differentiation. Typically, in the WT situation, virtually all cells had aggregated to form a single, sometimes two, spherical EBs with sharp boundaries per drop (Fig 5A). At day 4, EBs appeared slightly larger and denser than at day 2 and also sometimes started to disaggregate. The morphology of EBs obtained from *Copg2*$^{-/-}$ cells was similar to WT EBs, indicating that γ2-COP is not necessary for EB formation (Fig 5B). In contrast, EBs obtained from *Copg1*$^{-/-}$ cells had a strikingly distinct morphology with much less well-defined shapes and boundaries, and a general looser and more scattered appearance. At day 4 of differentiation, the difference was even more striking with hardly recognizable tight EBs and many disaggregated cells in the drops (Fig 5C). One possible explanation for this phenotype might be that in *Copg1*$^{-/-}$ cells, cell adhesion proteins are less efficiently trafficked to the cell surface.

These observations suggest that expression of γ1-COP is necessary for EB formation. However, as the cell line used in these experiments was obtained from a clonal selection after a Cas9-induced double-strand break within the *Copg1* gene, we decided to perform additional experiments aimed at addressing potential off-target effects of the sgRNA or clone-specific phenotypes. First, we constructed another *Copg1* KO cell line using a different strategy. Instead of inducing the formation of indels within the *Copg1* gene, the whole *Copg1* locus was removed by two single cuts induced by the Cas12a enzyme guided by two CRISPR RNAs (crRNAs) and replaced by the coding sequence of either GFP (P19 GFP KI cell line) or of γ1-COP-GFP (P19 γ1-COP-GFP KI cell line) as a control (Fig S3A and B, and see the Materials and Methods section). In these KI cell lines, disruption of the *Copg1* locus (GFP KI) also led to strongly impaired EB formation, whereas replacement of the *Copg1* locus by the coding sequence of γ1-COP-GFP had no effect (Fig 5D and E). The fact that both the *Copg1*$^{-/-}$ and GFP KI cell lines show similar phenotypes but are two different clones derived from two different

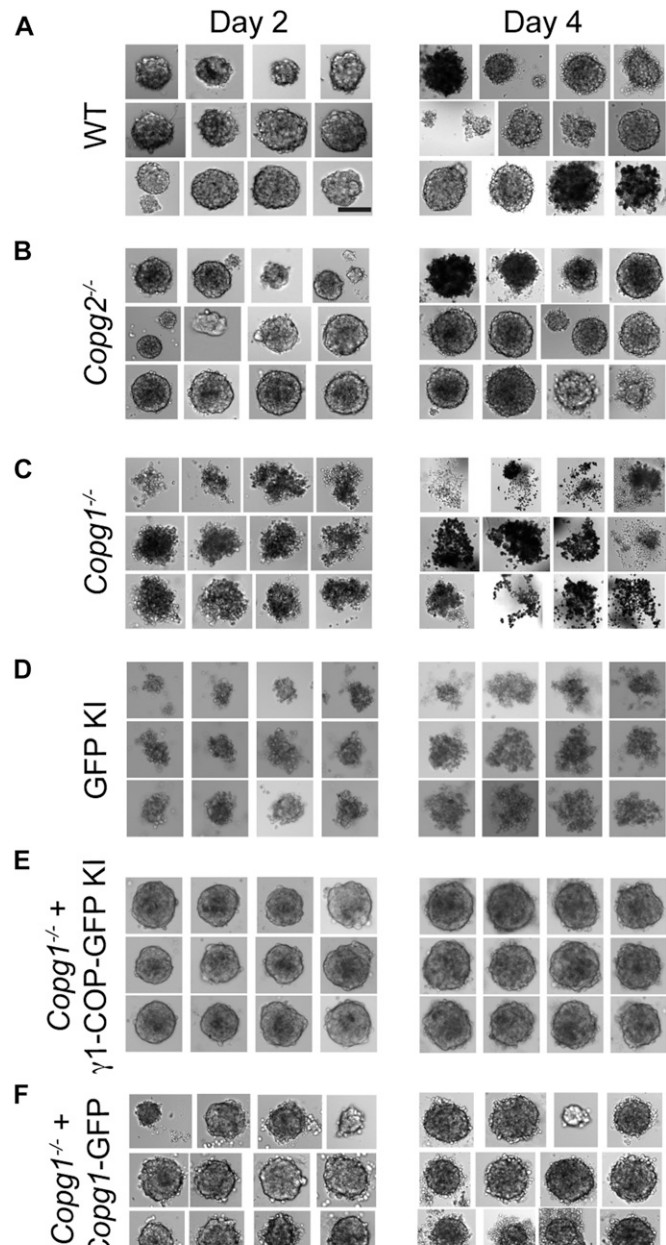

**Figure 5. Depletion of γ1-COP affects the formation of embryoid bodies.**
**(A, B, C, D, E, F)** Photographs of embryoid bodies from P19 WT (A), *Copg2*$^{-/-}$ (B), *Copg1*$^{-/-}$ (C), GFP KI (D), *Copg1*$^{-/-}$ + γ1-GFP KI (E), and *Copg1*$^{-/-}$ + *Copg1*-GFP (F) cells formed in hanging drops over 2 or 4 d of culture as indicated. **(A)** Scale bar in (A) is 50 µm and applies to all panels.

genome editing strategies strongly suggests that the absence of γ1-COP is the cause of defective EB formation.

As a further specificity control, we followed a rescue approach in which γ1-COP was reintroduced into *Copg1*$^{-/-}$ cells. We first constructed rescued cell lines through the transfection of *Copg1* and *Copg2* KO cells with a PiggyBac (PB) transposon vector driving the constitutive expression of γ1-COP and γ2-COP, respectively, through a strong CAG promoter (resulting in *Copg1*$^{-/-}$-PB-*Copg1* and *Copg2*$^{-/-}$-PB-*Copg2* cell lines). However, western blot analysis revealed that strong

overexpression of one paralog led to a much weaker expression of the other paralog (Fig S4A). In fact, *Copg1*$^{-/-}$-PB-*Copg1* cells appeared like a mimic of *Copg2* KO cells, whereas *Copg2*$^{-/-}$-PB-*Copg2* cells looked like a mimic of *Copg1* KO cells (Fig S4A, compare lanes 3–5 and 6–2). Previous studies showed that, except for ζ-COP, individual COP subunits are not detectable as monomers in the cytoplasm and only exist as part of the heptameric coatomer complex (Kuge et al, 1993; Lowe & Kreis, 1996). We therefore assume that the overexpressed γ-COP efficiently competes with its endogenous counterpart for assembly in the coatomer complex and that the excessive non-assembled protein gets rapidly degraded. As our goal was to obtain a rescue cell line with both γ-COP paralogs expressed close to their endogenous concentrations, we then turned to another strategy in which we used a bacterial artificial chromosome (BAC) that carries the whole *Copg1* gene locus under the control of its native promoter. Through recombineering, the *Copg1* gene was fused at the 3′ end to the localization and affinity purification tag that comprises a GFP sequence (Cheeseman & Desai, 2005). This strategy ensures that the reintroduced transgene is expressed at levels similar to the endogenous gene and under the control of its native regulatory elements (Poser et al, 2008). Western blot analysis of lysates of the rescue cell line (termed P19 *Copg1*$^{-/-}$ + *Copg1*-GFP) revealed that γ1-COP-GFP was indeed expressed at similar levels to endogenous γ1-COP in the WT cell line (Fig S4A, lane 4). Moreover, by disrupting *Copg2* in this rescued cell line, we successfully obtained P19 cells that solely express γ1-COP-GFP in a double *Copg1*$^{-/-}$, *Copg2*$^{-/-}$ background (Fig S4B). As the COPI pathway is essential, this demonstrates that GFP-tagged γ1-COP is functional. Accordingly, the P19 *Copg1*$^{-/-}$ + *Copg1*–GFP–rescued cell line showed faster proliferation rates than *Copg1*$^{-/-}$ cells (Fig S4C) and formation of tight EBs (Fig 5F). Hence, altogether, our data show that it is indeed the absence of the γ1-COP protein that prevents proper formation of EBs during neuronal differentiation of *Copg1* KO cells.

## Expression of γ1-COP promotes neurite outgrowth

We then analyzed WT and KO cells during the second step of neuronal differentiation, in which EB cells are dissociated, seeded on poly-lysine–coated plates (day 5 of differentiation) and grown for an additional 4 d (until day 8 of differentiation). 2 d after plating, cytosine arabinoside (ara-C) was added to poison dividing cells and hence enrich for post-mitotic neurons. During the second stage, P19 WT cells efficiently differentiated into neurons with the strong up-regulation of βIII-tubulin and the extension of long neurites (Fig 6A, top row). P19 *Copg2*$^{-/-}$ cells showed the same hallmarks of neuronal differentiation as WT cells with increased βIII-tubulin expression and long neurites (Fig 6A, second row from top). By contrast, *Copg1*$^{-/-}$ cells hardly showed any long neurites at day 8 of differentiation even though the cells expressed βIII-tubulin (Fig 6A, third row from top). The P19 *Copg1*$^{-/-}$ + *Copg1*–GFP–rescued cell line showed similar neurite extensions to WT and *Copg2*$^{-/-}$ cells (Fig 6A, bottom row). Further demonstrating the specificity of the *Copg1*$^{-/-}$ cell phenotype, the GFP KI cell line also hardly showed any extended neurites, whereas the γ1-COP-GFP KI cell line behaved similar to the WT cells (Fig S5). Semi-supervised automated measurements of neurite lengths from WT, KO, and rescued cells were performed using the NeuriteQuant software (Dehmelt et al, 2011). The quantification

corroborated the qualitative observations with a similar higher average neurite length per cell for WT, *Copg2*$^{-/-}$, and rescued *Copg1*$^{-/-}$ cells than *Copg1*$^{-/-}$ cells (Fig 6B). We did not measure a significant difference of the number of neurites per Tuj-1–positive cells between the four cell lines (Fig S6A); however, with cells such as P19 that grow at relatively high densities, the measurement of this parameter might be less reliable because the assignment of unique neurite attachment points is challenging.

When performing these assays, we reproducibly observed fewer remaining cells on the dishes at day 8 of differentiation when using *Copg1*$^{-/-}$ or GFP KI cells, suggesting that these cells differentiate less efficiently. Indeed, if the γ1-COP–lacking cells yielded fewer post-mitotic neurons, they would be more sensitive to the Ara-C treatment, which is toxic to dividing cells. To explore this possibility, we performed the differentiation protocol in the presence or absence of Ara-C for P19 WT, *Copg1*$^{-/-}$, and GFP KI cell lines. For each cell line, we then analyzed the percentage of recovered cells after Ara-C treatment (Fig S7A). We found that both γ1-COP–lacking cell lines were more sensitive to Ara-C than WT cells, indirectly suggesting that they differentiate less efficiently into post-mitotic neurons. In support to this hypothesis, immunofluorescence microscopy analysis at day 8 of differentiation showed much less cells positive for the neuronal marker Tuj-1 in the absence of Ara-C treatment for *Copg1*$^{-/-}$ and GFP KI cells than in the WT situation (Fig S7D). Importantly, *Copg1*$^{-/-}$ and GFP KI cells did not completely fail to differentiate. Indeed, similar to WT and *Copg2*$^{-/-}$ cells, *Copg1*$^{-/-}$ cells rapidly lost expression of Oct-4 (Fig 6C), indicating loss of pluripotency. Moreover, the remaining *Copg1*$^{-/-}$ and GFP KI cells after the Ara-C treatment showed expression of βIII-tubulin (Figs 6A and C and S5), implying a correctly implemented neuronal differentiation program.

Altogether, the data suggest that expression of γ1-COP regulates the efficiency of neuronal differentiation and promotes neurite outgrowth.

## Generation of P19 cells expressing γ2-COP from the *Copg1* locus

In non-differentiated *Copg1*$^{-/-}$ cells, the absence of γ1-COP is at least partly compensated by an increased expression of γ2-COP that seems to maintain global coatomer levels (Fig 2A–D). However, as we observed an increase in γ1-COP expression at the expense of γ2-COP in neurons derived from P19 WT cells (Fig 1), it is possible that *Copg1*$^{-/-}$ cells do not express enough γ2-COP to compensate at the neuronal stage, thereby restricting the amount of assembled coatomer, and possibly leading to the impaired neurite outgrowth phenotype.

Quantification by western blot of γ2-COP expression in *Copg1*$^{-/-}$ cells at the non-differentiated, EB, and neuron stages indicates that in contrast to WT cells, *Copg1*$^{-/-}$ cells do show an increased expression of γ2-COP at the neuron stage (Figs 6C and S8). However, as previous estimations found that there is about twice as much γ1-COP than γ2-COP in various cell lines (Wegmann et al, 2004; Moelleken et al, 2007), it is possible that the increased expression of γ2-COP in *Copg1*$^{-/-}$ cells is not sufficient to maintain physiological levels of γ-COP in P19 cells throughout differentiation. In that case, we cannot exclude that in *Copg1*$^{-/-}$ cells, a substantial portion of the cellular coatomer simply misses the γ-COP subunit, as was previously suggested in β-COP–depleted cells (Styers et al, 2008). Our immunoprecipitation data (Fig 2C) cannot address this possibility as we do not know how the western blot signals for γ1- and

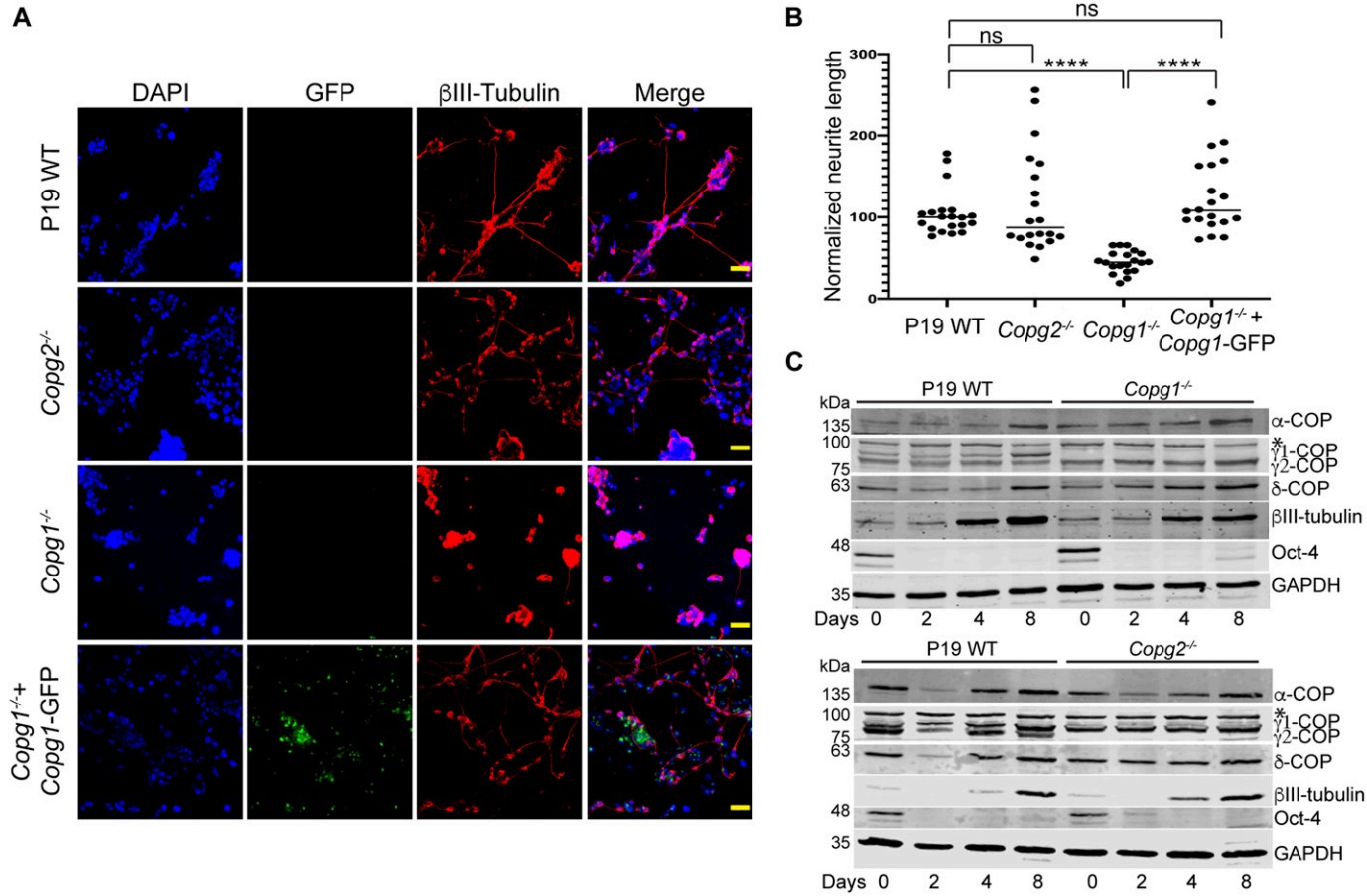

**Figure 6. Depletion of γ1-COP leads to impaired neurite outgrowth.**
**(A)** Representative fluorescence microscopy images of P19 WT, *Copg2*$^{-/-}$, *Copg1*$^{-/-}$, and *Copg1*$^{-/-}$ + *Copg1*-GFP cells as indicated at day 8 of differentiation to analyze the expression of the neuronal marker βIII-tubulin (indirect immunofluorescence) and GFP (direct fluorescence). **(B)** Quantification of the normalized neurite length (number of pixel/cell). **(A)** Each dot represents the value obtained for one picture (usually corresponding to ca. 100 cells) as in (A). 20 random images (a total of ca. 2,000 cells) were taken for the analysis. The horizontal bars represent the median value (set to 100 for WT) obtained for each cell line. **** indicates *P*-value < 0.0001, n = 20. Scale bar is 60 μm. **(C)** Western blot analysis of various COP subunits, βIII-tubulin, Oct-4, and GAPDH in P19 WT, *Copg1*$^{-/-}$, and *Copg2*$^{-/-}$ cells at different time point of differentiation as indicated. The asterisk (*) marks a nonspecific signal.
Source data are available for this figure.

γ2-COP correlate with the actual concentrations of the proteins. Hence, to characterize the capacity of γ2-COP to replace γ1-COP in a more defined way, we decided to substitute the γ2-COP coding sequence for γ1-COP at the endogenous *Copg1* locus. We thus generated a P19 γ2-COP-GFP KI cell line following the same Cas12a-mediated genome editing strategy that was used to generate the P19 GFP KI and P19 γ1-COP-GFP KI cell lines (Fig S3A and B).

**Expression of γ2-COP from the *Copg1* locus or overexpression of γ2-COP rescues the formation of tight EBs**

We next assessed if EB formation specifically requires the γ1-COP protein or just the expression of enough γ-COP irrespective of the paralog identity. To do so, P19 WT, GFP KI, γ1-COP-GFP KI, and γ2-COP-GFP KI cell lines were used to form EBs in hanging drops. As shown above, whereas WT and γ1-COP-GFP KI formed tight EBs, GFP KI cells showed scattered EBs (Fig 7A). Strikingly, γ2-COP-GFP KI cells were able to form tight EBs similar to WT cells (Fig 7A, bottom rows).

Because these cells do not express γ1-COP, this demonstrates that the phenotype of *Copg1*$^{-/-}$ and GFP KI cells is not due to a missing specific function mediated by γ1-COP. It rather appears that the absence of γ1-COP is not fully compensated by the up-regulation of endogenous γ2-COP in *Copg1*$^{-/-}$ cells, but that otherwise both paralogs can support the formation of EBs. To further explore this possibility, we turned to the PB-rescued cell lines we initially generated to rescue *Copg1* and *Copg2* KO cells. Indeed, in these two cell lines, overexpression of one or the other γ-COP paralog leads to essentially one population of coatomer complex that contains the overexpressed protein (Fig S4A). Strikingly, both rescue cell lines showed EB formation comparable with WT cells (Fig 7B). Whereas this was expected for *Copg1*$^{-/-}$-PB-*Copg1* cells as they mimic *Copg2* KO cells, the phenotype of *Copg2*$^{-/-}$-PB-*Copg2* cells suggests that when enough γ2-COP is expressed, EB formation can proceed normally even in the absence of γ1-COP. Thus, our data show that rather than the presence of a specific γ-COP paralog, it is a sufficient expression level of γ-COP that is essential for the formation of EBs in P19 cells.

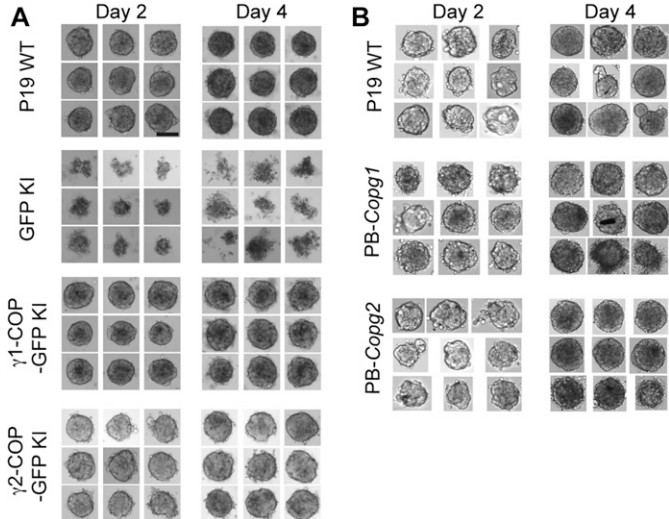

**Figure 7. Overexpression of γ2-COP rescues the formation of tight embryoid bodies.**
**(A)** Photographs of embryoid bodies from P19 WT, GFP-KI, *Copg1*⁻/⁻ + γ1-COP-GFP KI, and *Copg1*⁻/⁻ + γ2-COP-GFP cells formed in hanging drops over 2 or 4 d of culture as indicated. **(A, B)** Same as in (A) with P19 WT, PB-*Copg1*, and PB-*Copg2* cells. **(A)** Scale bar in (A) is 50 μm and applies to all panels.

### Expression of γ2-COP from the *Copg1* locus or overexpression of γ2-COP does not compensate for the absence of γ1-COP during neurite outgrowth

We next analyzed if the absence of γ1-COP can also be compensated by overexpression of γ2-COP during the second step of in vitro neuronal differentiation during which neurite outgrowth occurs. As shown above (Fig S5), microscopic analysis revealed that, whereas P19 GFP KI cells fail to extend long neurites, P19 γ1-COP-GFP KI cells show a network of long neurites comparable with WT cells (Fig 8A). In addition, P19 γ1-COP-GFP KI cells are more resistant to Ara-C treatment than P19 GFP KI cells (Fig S7B). Interestingly, expressing γ2-COP-GFP from the *Copg1* locus also resulted in a

resistance to Ara-C comparable with WT cells (Fig S7C), suggesting an increased differentiation efficiency compared with γ1-COP–lacking cells. Moreover, P19 γ2-COP-GFP KI cells could extend longer neurites than γ1-COP–lacking cell lines; however, their lengths and amount were not comparable with those of WT cells (Figs 8 and S6B). These observations were corroborated by the software-mediated quantification that showed a significantly lower average neurite length per cell for the γ2-COP-GFP KI cells when compared with WT and γ1-COP-GFP KI cells (Fig 8B). Hence, in contrast to the EB formation step, it appears that γ1-COP has a specific role during the formation of extended neurites. This was corroborated by the PB-rescued cell lines. Indeed, overexpression of γ1-COP in *Copg1*⁻/⁻ cells (PB-*Copg1* cells) resulted in the formation of neurons with long neurites comparable with the WT situation. In contrast, overexpression of γ2-COP in *Copg2*⁻/⁻ cells (PB-*Copg2* cells) led to the formation of neurons, but with much shorter neurites (Fig S9A and B). Hence, replacement of γ1-COP by an excess of γ2-COP induces a phenotype comparable with the disruption of the *Copg1* gene, suggesting that γ1-COP has a unique role during neuronal differentiation that cannot be taken over by γ2-COP. Of note, as *Copg2*⁻/⁻-PB-*Copg2* cells form normal EBs but fail to extend long neurites (Fig 7B); the defective neurite outgrowth observed in *Copg1* KO cells is unlikely to be a mere consequence of their incapacity to form proper EBs.

Altogether, our data reveal a paralog-specific function of the COPI pathway during the neuronal differentiation of P19 cells. In this context, γ1-COP promotes efficient neurite outgrowth in differentiated neurons.

## Discussion

The *Copg2* and *Copz2* genes, paralogous to *Copg1* and *Copz1*, were discovered two decades ago (Blagitko et al, 1999; Futatsumori et al, 2000). Since then, it has been an outstanding question whether unique functions can be ascribed to COPI paralogous proteins (Béthune et al, 2006; Arakel & Schwappach, 2018; Luo & Boyce, 2019). Recombinant coatomer complexes that contain either γ1-COP or

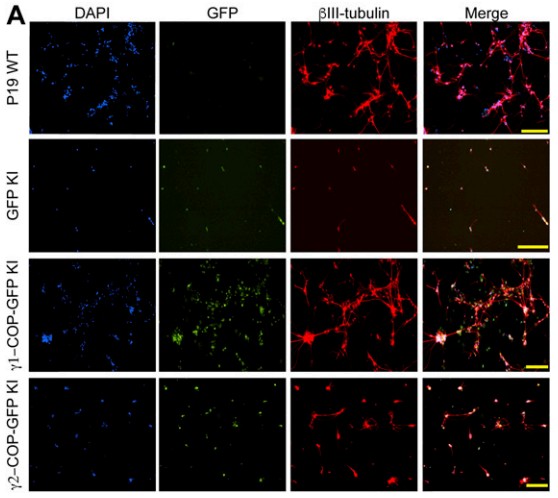

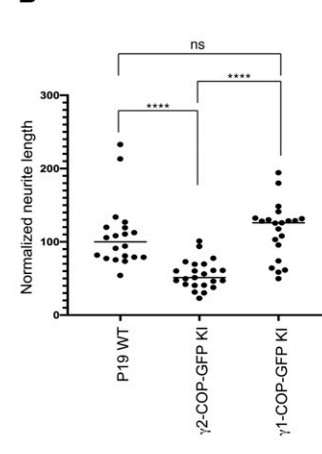

**Figure 8. Overexpression of γ2-COP does not compensate the absence of γ1-COP during neurite outgrowth.**
**(A)** Representative fluorescence microscopy images of P19 WT, GFP KI, γ1-COP-GFP KI, and γ2-COP-GFP KI cells as indicated at day 8 of differentiation to analyze the expression of the neuronal marker βIII-tubulin (indirect immunofluorescence) and GFP (direct fluorescence). Scale bar is 200 μm. **(B)** Quantification of the normalized neurite length (number of pixel/cell). **(A)** Each dot represents the value obtained for one picture (usually corresponding to ca. 100 cells) as in (A). 20 random images (a total of ca. 2,000 cells) were taken for the analysis. The horizontal bars represent the median value (set to 100 for WT) obtained for each cell line. **** indicates *P*-value < 0.0001, ns, nonsignificant, n = 20.

γ2-COP are equally efficient in producing COPI vesicles from purified Golgi membranes (Sahlmuller et al, 2011) or permeabilized cells (Adolf et al, 2019). Moreover, the proteomes of COPI vesicles prepared with γ1-COP– or γ2-COP–containing coatomer and using permeabilized HeLa cells as donor membranes were virtually indistinguishable (Adolf et al, 2019). Hence, these previous in vitro data suggested that γ1-COP and γ2-COP are functionally redundant. Our present study performed with living cells revisits this conclusion by showing that whereas γ1-COP and γ2-COP are indeed partially functionally redundant, a paralog-specific function of γ1-COP promotes the extension of neurites in pluripotent cell-derived neurons.

The COPI pathway is essential for life, and depletion of non-paralogous COP subunits is lethal in mammalian cells (Hobbie et al, 1994; Shtutman et al, 2011). As we obtained cell lines that lack either γ1-COP or γ2-COP, our data demonstrate that both proteins can support the essential functions of the COPI pathway and that none of the two γ-COP paralogs is essential. However, both *Copg1* and *Copg2* KO cell lines had slower proliferation rates than WT cells, indicating that γ1-COP and γ2-COP have some nonessential unique functions. Supporting this hypothesis, whereas disruption of *Copg1* or *Copg2* did not lead to a complete loss of the Golgi as reported upon the depletion of the non-paralogous subunits α- or β-COP (Razi et al, 2009), depletion of either γ1-COP or γ2-COP led to smaller and more numerous Golgi stacks.

Rapid evolution of dosage sharing is a main driving force to maintain gene duplicates in mammals (Lan & Pritchard, 2016). Hence, a difficulty when assessing individual KO of paralogous gene pairs is to disentangle whether the cause of a phenotype is a unique function that can be assigned to one paralog or an aberrant combined expression level of two paralogous genes with essentially overlapping functions. Indeed, whereas our initial data suggested a role of γ1-COP in promoting the formation of tight EBs, KI, and overexpression experiments clearly showed that γ2-COP can take over γ1-COP's role when expressed in sufficient quantity. This suggests that in this context, expression of γ1-COP and γ2-COP is regulated by dosage balance to achieve a suitable combined expression of γ-COP compatible with a functional COPI pathway. By contrast, the same combination of experiments supports a unique role for γ1-COP that cannot be taken over by γ2-COP in promoting efficient neurite extension at a later stage of P19 cell neuronal differentiation. This finding correlates with the initial observation that in WT cells, expression of γ1-COP is up-regulated at the expense of γ2-COP only at later stages of differentiation, when neurite outgrowth occurs.

What could be the molecular mechanism underlying the role of γ1-COP in promoting neurite outgrowth? Previous studies already hinted at an important role of ER/Golgi trafficking during neuronal polarization. For example, overexpression of the GTP-locked Q71L-ARF1 variant or treatment with the coatomer complex inhibitor 1,3-cyclohexanebis(methylamine) (Hu et al, 1999) leads to impaired dendritic growth in cultured hippocampal neurons (Horton et al, 2005). Moreover, mutations in the COPII components Sec23 and Sar1 also lead to reduced dendritic growth in flies (Ye et al, 2007). In these examples, the outcome of the mutations or treatment is a virtually complete block of ER/Golgi trafficking. This is, however, not expected in the absence of γ1-COP, especially with the concomitant overexpression of γ2-COP, as γ1-COP–depleted cells are viable and γ2-COP–containing coatomer is as efficient in forming COPI vesicles as γ1-COP–containing coatomer in vitro (Sahlmuller et al, 2011; Adolf et al, 2019). One possibility is that γ1-COP acts as a specific cargo receptor for proteins that are important for neuronal polarization. In the absence of γ1-COP, such hypothetical cargo proteins may be mislocalized, leading to the phenotype of *Copg1* KO cells. The proteomes of COPI vesicles generated with γ1-COP– or γ2-COP–containing coatomer were reported as virtually identical (Adolf et al, 2019). However, these vesicles were generated from HeLa cells and putative γ1-COP–specific cargo proteins might be neuron specific. The γ-COP paralogs show a differential Golgi localization with γ1-COP being found preferentially at the *cis*-Golgi and γ2-COP more at the *trans*-Golgi (Moelleken et al, 2007). This suggests that γ-COP paralogs help maintaining protein concentration gradients across the Golgi stack, with γ1-COP being more active at the ER/*cis*-Golgi interface. Proteomic analysis of COPI vesicles generated from various cell lines revealed that they mainly contain membrane trafficking regulating proteins, such as SNAREs, and glycosylation enzymes (Adolf et al, 2019). The absence of one γ-COP paralog may then affect the distribution or trafficking kinetics of such proteins across the Golgi, which might explain the phenotype of *Copg1* KO cells. Indeed, mutations in both SNAREs (Ulloa et al, 2018) and glycosylation enzymes (Moll et al, 2019) are linked to defective neurite outgrowth and neurological disorders. Notably, within the SNARE protein Membrin, pathogenic mutations that lead to less effective fusion of ER-derived vesicles with the Golgi, result in strongly impaired dendritic growth but do not significantly affect non-neuronal cells (Praschberger et al, 2017). This phenotype, which is similar to the depletion of γ1-COP, highlights the requirement of an intact early secretory pathway for neurite outgrowth. As we observed that depletion of γ1-COP results in a more drastic Golgi fragmentation phenotype than depletion of γ2-COP, the two γ-COP paralogs might affect the distribution of Golgi proteins differently, which might explain why only γ1-COP is required for neurite outgrowth.

Finding the proteins that are mislocalized upon depletion of γ1-COP will be key to understanding the molecular mechanisms underlying γ1-COP's function during neuronal polarization. This is at present not possible with a direct proteomic analysis of isolated COPI vesicles from neurons, considering the amount of starting material needed and the limit of detection of the assay (Adolf et al, 2019). More promising might be more global approaches such as dynamic organellar maps (Itzhak et al, 2016) or Hyperplexed Localisation of Organelle Proteins by Isotope Tagging (hyperLOPIT) (Christoforou et al, 2016) in which changes in protein localization upon γ1-COP depletion may be analyzed at the proteome level.

Altogether, we found that whereas both γ-COP paralogs can fulfill essential coatomer functions, specialized paralog-specific functions exist. We reveal here a role for γ1-COP in promoting neurite outgrowth, and it is likely that in other cellular contexts, additional paralog-specific functions of COP subunits await to be found. Of note, many cancer cell lines are sensitive to a down-regulation of ζ1-COP because they barely express its paralog ζ2-COP (Shtutman et al, 2011; Anania et al, 2015, 2017). Potential drugs that would target ζ1-COP have thus been proposed as attractive therapeutic options with supposedly limited side effects because

ζ2-COP is expressed in normal cells (Shtutman & Roninson, 2011; Anania et al, 2017). Our study calls for caution when following such approaches as ζ1-COP, like γ1-COP, may have important functions that cannot be taken over by its paralog in specific cellular contexts.

# Materials and Methods

## Antibodies and reagents

Reagents, plasmids, antibodies, and primers used in this study are listed in Tables S1–S5 (materials used in this study, list of sgRNAs/crRNAs, plasmids used in this study, antibodies used in this study, list of primers used in qPCR experiments). Quantification of western blots was performed with the LI-COR Image Studio software according to the distributor's instructions.

## Cell culture

The P19 cell line was obtained from Sigma-Aldrich and kept at 37°C with 5% $CO_2$ in a humidified incubator. The growth medium was Minimum Essential Medium Eagle (α-MEM) supplemented with 10% FBS (S181B; Biowest), 2 mM glutamine, and a mixture of penicillin/streptomycin. All the cell lines were regularly checked for mycoplasma contamination.

## P19 differentiation

P19 cells were differentiated into neurons using a two-step protocol essentially as previously described (Lee et al, 2007). First, $10^6$ low passage P19 cells were used to seed a 10-cm bacterial dish containing 10 ml of P19 growth medium supplemented with 0.1 $\mu$M RA. These non-adherent growth conditions promote cell aggregation and the formation of EBs. After 48 h of aggregation, the medium was replaced with fresh RA-supplemented P19 growth medium. After 4 d of aggregation, EBs were centrifuged at 1,000$g$ for 5 min and then washed once with serum-free media. Then EBs were dissociated using 2 ml of a trypsin (0.05%)–EDTA (0.02%) solution (Cat. no. T3924; Sigma-Aldrich) + 50 $\mu$g/ml DNAseI and incubated for 10 min in a 37°C incubator. Next, 4 ml of P19 growth medium were added to stop the trypsin activity and the cells are collected at 1,000$g$ for 5 min. Cell pellets were then resuspended in 5 ml growth medium and passed through a cell strainer (352235; Falcon). Dissociated cells were used to seed poly-L/D-lysine–coated six-well plates (7.5 × $10^5$ cells in 2 ml per well) or, for fluorescence microscopy experiments, to seed poly-D-lysine–coated eight-well chambered coverslip (80826; Ibidi) with 40,000 cells per well in 300 $\mu$l P19 growth medium. 48 h after plating (at day 6 of differentiation), the medium was exchanged for P19 growth medium supplemented with 10 $\mu$M Ara-C to poison dividing cells. A list of chemicals and their suppliers is provided in Table S1.

## Generation of the P19 $Copg1^{-/-}$ and $Copg2^{-/-}$ KO cell lines

P19 $Copg1^{-/-}$ and $Copg2^{-/-}$ cells were generated by following a CRISPR-Cas9 gene editing strategy. sgRNAs designed to cut in early

exons of the $Copg1$ and $Copg2$ genes (selected with the CRISPR.MIT tool, see Table S2) were cloned into the pSPCas9(BB)-2TA-GFP vector between the BbsI restriction sites (Ran et al, 2013). P19 cells were transfected with these vectors using Lipofectamine 3000 (Thermo Fisher Scientific) following the manufacturer's instructions. 72 h after transfection, single GFP-positive cells were sorted and transferred into a 96-well plate using a FACS at the CellNetworks/ZMBH-Flow Cytometry & FACS Core Facility. Clonal cell lines were grown and transferred into larger cell culture dishes and then screened by western blot. Clones that showed absence of γ1-COP or γ2-COP were verified by pCR-Blunt cloning/Sanger sequencing of the target locus.

## Generation of P19 $Copg1^{-/-}$-PB-$Copg1$ and $Copg2^{-/-}$-PB-$Copg2$ cell lines

The P19 $Copg1^{-/-}$-PB-$Copg1$ and $Copg2^{-/-}$-PB-$Copg2$ cell lines were generated using the PB transposon system (Wang et al, 2008). P19 $Copg1^{-/-}$ and $Copg2^{-/-}$ cells seeded in a six-well plate were transfected with an equal amount (1.5 $\mu$g each) of pPbase plasmid (coding for the PB transposase) and pCyl50-$Copg1$ or pCyl50-$Copg2$ plasmid (coding for the gene to be inserted and the hygromycin resistance gene). In addition, a third control cell line was generated by co-transfection of pPbase and a pCyl50-GFP plasmid. 24 h after transfection, the cells were transferred to a 10-cm plate. After 3 d of incubation, hygromycin was added to the growth medium (at 150 $\mu$g/ml), and the medium was replenished every second–third day. After 15 d, all GFP-transfected control cells were green as judged by fluorescence microscopy. Transfected PB-$Copg1$ and PB-$Copg2$ cells were kept on selection media for another 5 d and then analyzed by western blot to check expression of γ1-COP or γ2-COP, respectively. Table S3 lists the plasmids used in this study.

## Generation of the P19 $Copg1^{-/-}$-$Copg1$-GFP cell line

BACs harboring the murine $Copg1$ locus were GFP tagged by recombineering as described previously (Poser et al, 2008). BAC DNA was isolated from Escherichia coli DH10 cells using a BAC prep kit (Macherey-Nagel). P19 $Copg1^{-/-}$ cells were transfected with 1 $\mu$g BAC DNA using Effectene (QIAGEN), and stable clonal cell lines were obtained after selection with 500 $\mu$g/ml geneticin (Gibco) and FACS sorting.

## Generation of P19 KI cells

P19 KI cells were generated by following a CRISPR-Cas12a/Cpf1 gene editing strategy including crRNAs arrays, a donor template for homology-directed repair (HDR) and a recombination reporter plasmid for selection. Two crRNAs (see Table S2) targeting the $Copg1$ gene within the 5'UTR and 3'UTR coding sequencing were selected with the CHOPCHOP tool (Labun et al, 2019) and cloned as a dual array between the BsmbI sites of pY109 (lenti-LbCpf1) plasmid (Zetsche et al, 2017) derivative (pY109_2) in which the Puro-selection cassette was removed. The donor template cassettes were PCR products containing homology arms (left 75 bp and right 55 bp for HDR) to the target locus flanking the coding sequence for either GFP (GFP KI cells), γ1-COP-GFP (γ1-COP-GFP KI cells), or γ2-COP-GFP (γ2-COP-GFP KI cells). The donor cassettes were

assembled using the NEBuilder HiFi DNA Assembly Master Mix (NEB) and were designed to leave the 5′UTR and 3′UTR of the *Copg1* gene intact. The recombination reporter plasmid (pMB1610_pRR-Puro) allows a split puromycin resistance approach for selecting cells in which Cas12a cut and HDR occurred (Flemr & Buhler, 2015). The target sequence for crRNA2 was cloned between the two puromycin fragments cassettes within pMB1610. When the Cas12a-induced DNA strand break in the plasmid is repaired via HDR this results in a functional puromycin resistance. P19 cells were seeded at 200,000 cells per well of a six-well plate. On the next day they were transfected with a mixture of 1.5 μg of modified pY109-crRNA array plasmid, 1 μg of PCR donor cassette and 0.5 μg of pMB1610 reporter plasmid using the JetPrime reagent (Polyplus-transfection) following the manufacturer's instructions. Puromycin was added 24 h later to 4 μg mL⁻¹ in the growth medium and the cells incubated for another 48 h. Surviving cells were then transferred to a 10 cm dish and allowed to grow to 70–80% confluence. A pool of GFP-positive cells was then selected by FACS and transferred to a 10 cm dish. The cells were allowed to grow to 70–80% confluence and then single GFP-positive cells were selected and transferred to 96-well plates by FACS. Clonal cell lines were grown and transferred to larger cell culture dishes. Cells that showed a Golgi-like GFP signal by fluorescence microscopy were kept for further screening. Suitable clones were selected after western blot analysis of γ1-COP and γ2-COP expression (absence of endogenous γ1-COP and presence of either γ1-COP-GFP or γ2-COP-GFP). The molecular nature of all genome-edited alleles was verified by PCR (outside homology arms to verify a correct junction of the homology arms with the transgenes, and between exon 8 and 9 of *Copg1* to verify the absence of the endogenous *Copg1* locus) and by pCR-blunt/Sanger sequencing of the complete modified target locus. With these controls we made sure that all three KI cell lines had a bi-allelic insertion of the transgenes.

### Hanging drop assay

P19 cells were diluted to 10,000 cells/ml in RA-containing P19 growth medium. 30 drops of 20 μl volume (containing 200 cells) were then deposited inside the lid of a 10-cm bacterial dish filled with 10 ml PBS to avoid evaporation. On the second and fourth day of EB formation, images of individual drops were taken on a Nikon TS100F microscope fitted with a CMOS camera (Imaging source, DMK 23UX174) using 10× air objective.

### Immunostaining

For immunostaining, cells were washed with preheated (37°C) PBS, then fixed with PBS supplemented with 4% formaldehyde for 20 min at 37°C. Fixed cells were washed twice with PBS, and then permeabilized with PBS + 0.25% Triton X-100 at RT for 10 min. Cells were then washed twice with PBS + 2% BSA and blocked with PBS + 10% BSA for 30 min at RT. Incubation with primary antibodies diluted in PBS + 2% BSA was then performed either for 1 h at RT or overnight at 4°C. Next, cells were washed three times with PBS + 2% BSA and incubated with the relevant secondary antibody diluted in PBS + 2% BSA for 30 min at RT in the dark. Cells were then washed twice with PBS + 2% BSA and incubated with DAPI (4′,6-diamidino-2-phenylindole diluted at 0.1 μg/ml in PBS) for 10 min at RT in the dark. After three brief and gentle washes, the first two with PBS +2%

BSA, the last one with PBS, cells were mounted with Ibidi mounting medium. Images were acquired with a Nikon ECLIPSE Ti2 microscope equipped with an Andor Clara DR or Nikon DS-Qi2 camera. Neurite length was assessed after βIII-tubulin staining by analysis with the NeuriteQuant software (Dehmelt et al, 2011). For each image, total neurite length was normalized to the number of nuclei counted with the Fiji software. A list of antibodies and their working dilutions is given in Table S4.

### Chemical treatment experiments

To induce apoptosis, P19 cells were treated with staurosporine (1 μM) for 20 h. Thereafter, immunostaining was performed (see above). To induce ER stress, P19 cells were treated with tunicamycin (2.5 μg mL⁻¹) for 24 h. Thereafter, western blot or FACS analysis was performed.

### Apoptosis flow cytometry assay

For each cell line, $300 \times 10^5$ cells were used to perform a FAM FLICA Caspase-3/7 assay (Cat. no. ICT093; Bio-Rad) according to the manufacturer's instructions. Measurements were performed on a BD FACSCanto flow cytometer. The recommended control samples were included to set PMT voltages and adjust fluorescent compensation.

### Real-time cell proliferation assay

To assess cell proliferation kinetics, 100,000 cells were seeded per well of a six-well plate and incubated in an IncuCyte Zoom live-cell analysis system (Essen Bioscience) within a 37°C incubator. Images were acquired in every 4 h for a period of 72 h. Data analysis was performed with the IncuCyte ZOOM Software.

### RNA isolation/cDNA preparation/RT-qPCR

RNA isolation, cDNA preparation, and RT-qPCR were performed exactly as described in Amaya Ramirez et al (2018). Primers used for qPCR are listed in Table S5.

### Cytosol preparation from adherent cells

For cytosol preparation, two confluent 15-cm cell culture dishes were used. Cells were washed twice with PBS and collected by scrapping in 0.5 ml of lysis buffer (25 mM Tris, pH 7.4, +150 mM NaCl +1 mM EDTA + Protease inhibitor cocktail). Scrapped cells were lysed by passing them 20 times (10 strokes) through a 21-gauge needle and then 20 times through a 27-gauge needle on ice. Cell lysates were first cleared at 4°C, 800*g*, 5 min and then at 4°C, 100,000*g*, 1 h. Protein concentration was estimated with a Bradford assay. Typically, a yield of ca. 1.5 mg protein per 15-cm plate was obtained.

### Coatomer immunoprecipitation from cytosolic preparations

Per IP, 10 μl Protein G magnetic beads (S1430S; NEB) were used. Beads were washed twice with IP buffer (25 mM Tris, pH 7.4 + 150 mM NaCl + 1 mM EDTA + 0.025% [vol/vol] tween 20). After the second wash, 100 μl of CM1 antibody supernatant was added per 10 μl Protein G beads, the tube was then filled to ca. 1.4 ml with IP buffer

and incubated for 1 h at RT on a rotating wheel. Then, the beads were washed once with IP buffer (1 ml). Freshly prepared cytosol corresponding to 500 µg protein was then added to the beads, and if needed, the tubes were filled to ca. 1.4 ml with IP buffer containing protease inhibitors. Cytosol and beads were incubated for 1 h at RT on a rotating wheel. The beads were then washed three times with IP buffer to remove unbound proteins. Bound proteins were eluted by incubating the beads with 20 µl 3× SDS loading buffer at 70°C for 10 min. Beads were then separated from the eluted material on a magnetic rack and the eluted material transferred to a fresh tube.

### Electron Microscopy

Cells in a six-well plate were fixed at RT for 30 min by adding the fixative solution (2.5% glutaraldehyde and 2% sucrose in 50 mM KCl, 2.6 mM MgCl$_2$, 2.6 mM CaCl$_2$, and 50 mM cacodylate buffer, pH 7.2). The cells were then rinsed five times for 2 min at RT with 0.1M cacodylate buffer, pH 7.2, and incubated at 4°C for 40 min with Contrast I solution (2% OsO$_4$ in 50 mM cacodylate buffer). The cells were then rinsed five times for 1 min at 4°C and then twice for 5 min at RT with mQ water. Thereafter, the cells were incubated at RT for 30 min in the dark with Contrast II solution (0.5% uranyl acetate in mQ water). Finally, the cells were dehydrated with increasing amounts of ethanol and embedded in epon epoxy resin (Polysciences). Ultrathin sections of 60 nm were contrasted with uranyl acetate and lead citrate using an AC20 automatic contrasting system (Leica) and examined with a T12 electron microscope (Thermo Fisher Scientific). Images were taken from WT, *Copg1*$^{-/-}$, and *Copg2*$^{-/-}$ cell lines. Images were collected from three different grids per condition. Random pictures of 30 cell profiles containing a Golgi stack were collected for quantification. All the measurements were done using ImageJ. For quantification of the area of individual Golgi stacks, the smallest possible ellipse was drawn around each Golgi stacks and the enclosed area was calculated. Cisternae lengths were measured by drawing a segmented line with ImageJ through cisternae for which clear beginnings and ends could be observed. Finally, the surface area of Golgi stacks versus Golgi-associated vesicles was assessed and the percentage of Golgi's that contained more than 50% vesicles was calculated. Average values and SEM were computed with the Prism 8 software.

### Statistical analysis

Statistical significances were calculated using a two-tailed unpaired Student's parametric *t* test with Welsh's correction with the Prism 8 software (GraphPad).

# Supplementary Information

# Acknowledgements

We thank Felix Wieland (Heidelberg) for the kind gifts of plasmids and antibodies (see Supplemental Data). We thank Ina Poser (Dresden) for BAC tagging and Menna Ahmed for her help in handling the IncuCyte device. We thank the CellNetworks EM, FACS, and Nikon imaging center facilities for their expert support. We thank F Wieland and Mandy Jeske for their critical comments on the manuscript. This work was supported by the excellence initiative of the German federal and state governments (DFG-EXC81).

## Author Contributions

M Goyal: formal analysis and investigation.
X Zhao: formal analysis and investigation.
M Bozhinova: investigation.
K Andrade-López: investigation.
C de Heus: formal analysis and investigation.
S Schulze-Dramac: investigation.
M Müller-McNicoll: formal analysis, supervision, funding acquisition, investigation, and writing—review and editing.
J Klumperman: formal analysis, supervision, funding acquisition, investigation, and writing—review and editing.
J Béthune: conceptualization, formal analysis, supervision, funding acquisition, investigation, project administration, and writing—original draft, review, and editing.

## Conflict of Interest Statement

The authors declare that they have no conflict of interest.

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
