## [Reviewer comments · Life Science Alliance]

Life Science Alliance

A paralog-specific role of COPI vesicles in the neuronal differentiation of mouse pluripotent cells

Manu Goyal, Xiyan Zhao, Mariya Bozhinova, Cecilia de Heus, Sandra Schulze-Dramac, Michaela Müller-McNicoll, Judith Klumperman, and Julien Béthune

DOI: <https://doi.org/10.26508/lsa.202000714>

Corresponding author(s): Julien Béthune, Heidelberg University

Review Timeline:

Submission Date:	2020-03-23
Editorial Decision:	2020-03-24
Revision Received:	2020-07-01
Editorial Decision:	2020-07-03
Revision Received:	2020-07-06
Accepted:	2020-07-06

Scientific Editor: Reilly Lorenz

Transaction Report:

Please note that the manuscript was previously reviewed at another journal and the reports were taken into account in the decision-making process at Life Science Alliance.

Reviewer #1 Review

Comments to the Authors (Required):

This manuscript describes analysis of the two different COPI gamma-COP subunit isoforms during neuronal differentiation. While it has been known for many years that two gamma-COP isoforms exist (gamma-1 and gamma-2 COP), the functional significance has remained mysterious. Here, using elegant genome editing approaches, it is shown that gamma-1 and gamma-2 COP are both required for optimum cell growth and Golgi structure, but cells remain viable upon loss of either. Analysis of neurogenesis in vitro shows that gamma-1 is more abundantly expressed than gamma-2 COP, and its loss disrupts formation of intermediate structures seen in the neurogenesis programme called embryoid bodies (EB), as well as neurite outgrowth that occurs at a later stage. Gamma-2 COP when more abundantly expressed can compensate the EB phenotype, but not neurite outgrowth, indicating a unique, paralog-specific role for gamma-1 COP in this process.

I found the results to be well presented, and convincing. The authors' conclusions were generally well supported by the data, and I found the study to be interesting. However, there are a couple of major points that lessen my enthusiasm for the work. The first is that the data are all generated using an in vitro system for neuronal differentiation. The study would be much stronger in my view if there was a demonstrable effect of gamma-1 COP manipulation in an in vivo model, where neurogenesis is occurring naturally. The second is that while the results clearly show gamma-1 COP is important for neurite outgrowth, albeit in vitro, there is a lack of mechanism to explain the phenotype (the same applies to the EB data). One might expect that disrupted trafficking is responsible, but this is not explored.

There are also some more minor specific points as detailed below:

- 1.) The EM analysis of Golgi morphology in Fig 3 could be more extensive. For example, are the number of cisternae per stack or cisternal length altered by loss of gamma-1 or gamma-2 COP? Counting Golgi stacks per cell seems very subjective, especially when it seems the Golgi area of the cell is pre-selected for counting. It seems odd that the numbers are so consistent when sectioning is likely to give a high degree of variability in terms of cutting through the Golgi stacks.
- 2.) The authors use BIP protein levels as a readout for ER stress (Fig 4A). This is quite an insensitive measure. Other analysis should be done to rule out an ER stress phenotype e.g. Xbp1 splicing.
- 3.) It is claimed that more gamma-2 COP KO cells show apoptosis than gamma-1 COP (Fig 4B). The microscopy images do not clearly show this effect, which should also be quantified.
- 4.) What does looser EB formation indicate? This phenotype is scored in several experiments but its significance is not obvious.

Reviewer #2 Review

Comments to the Authors (Required):

COPI coated vesicles are critical for traffic between Golgi cisternae, a function that is required in all cells. This manuscript investigates the possibility that the paralogous gamma1 and gamma2 subunits of the COPI coat may have distinct functions through a rigorous set of genetic perturbations and rescue experiments in mouse P19 carcinoma cells as well as in neuron-like cells derived from them. Through such efforts, the authors show that although either gamma-subunit can support viability, gamma1 is selectively required to support neurite outgrowth during differentiation of the neuron like cells. Although the authors present data that makes a compelling argument in favor of a distinct function for gamma1, insight into how gamma1 is able to support neurite outgrowth is lacking.

Some concern also arises about the generalizability of key observations outside of the specific model system examined. Although the differentiation model to convert P19 carcinoma cells into neuron-like cells yields cells with many properties of neurons, it is not clear that the observations from this one model system can be generalized to make broad statements about the requirement for gamma1 in neuronal development. Thus, although the manuscript shows a neurite outgrowth defect in the absence of gamma1, it remains unclear to what extent observations from this one neuron-like model system can be generalized to neurons in vivo. Nonetheless, the authors

frequently make statement that indicate that their discoveries will apply much more generally.

Reviewer #3 Review

Comments to the Authors (Required):

Comments to authors:

Overall, this is a strong manuscript showing that COPG1 is required for differentiation and neurite outgrowth in P19 cells. However, the manuscript lacks quantitative rigor, and this must be addressed before I can recommend publication.

Figure 1b) Why are only some of the COPI coatomer subunits measured in the cell lysates? Later figures show that the lab has a clean antibody to COPE, and commercial antibodies are available for both B and B'. The levels of all COPI coatomer members should be documented during differentiation. If Western blot analysis is going to be used to measure increase or decrease in proteins of interest, they blots should be analyzed by densitometry (compared to a suitable housekeeping protein such as GAPDH) and the levels of the proteins from multiple gels reported with appropriate statistical methods applied. This criticism applies to all blots presented in this paper.

Figure 2b) this blot shows that they are able to detect COPE by Western Blot, therefore COPE should be analyzed in figure 1 as well.

Figure 4a) Again, quantification of the Western blot is required. By eye, this particular blot appears to show increased GRP-78 in the COPG2^{-/-} lines compared to P19WT controls. The more appropriate experiment might be to treat both WT and COPG1^{-/-} and COPG2^{-/-} cultures with Tunicamycin to evaluate the percent increase in Grp-78 as it's unclear here whether the mutant lines are ABLE to respond to Tunicamycin at all.

Figure 4b) These results must be quantified. The authors state that "only occasional" caspase 3 staining was detected in WT cultures and "slightly more" was detected in COPG2^{-/-}. This vague, qualitative language is insufficient. Please provide concrete numbers from repeated experiments with appropriate measures of statistical significance.

Figure 5) I would suggest including the results presented in 7a as figure 5e - as a reader, I was confused as to why this experiment wasn't performed and by the time it was presented, it felt out of place.

Figure 6) the number of neurites per cell should be quantified as well as the average neurite length. The authors mention neurite number when discussing figure 8, but never present quantitation. 6c/d should be quantified by densitometry to verify statement of increased protein levels. The COPG2^{-/-} cultures appear to adopt B-III tubulin positive cells earlier than P19wt cultures, but this is not addressed. Is this finding consistent? Paired with the finding in 6b that the copg2^{-/-} cultures appear to have a trend towards longer neurites, this needs to be discussed.

When discussing the results of figure 6, the authors state that in CP{G^{-/-} cells a substantial portion of the cellular coatomer simply lacks COPG. This should be addressed using the B'-COP co-IP reported in figure 2C using lysate from differentiated P19 cultures.

Figure 8) The text declares "more and longer neurites" but only neurite length is quantified and presented. Please quantify and present neurite number with appropriate statistical methods to determine significance. This criticism applies to figure 6 as well.

March 24, 2020

Re: Life Science Alliance manuscript #LSA-2020-00714-T

Dr. Julien L Béthune
Heidelberg University
Heidelberg University Biochemistry Center (BZH)
Im Neuenheimer Feld 328
Heidelberg, Baden-Wuerttemberg 69120
Germany

Dear Dr. Béthune,

Thank you for transferring your manuscript entitled "A paralog-specific role of the COPI pathway in the neuronal differentiation of murine pluripotent cells" to Life Science Alliance. The manuscript was assessed by expert reviewers at another journal before, and the editors transferred those reports to us with your permission.

The reviewers who evaluated your work elsewhere thought that your data are robust. However, they would have expected further reaching mechanistic insight and in vivo relevance. Lack thereof does not preclude publication here, and we would thus like to invite you to submit a revised version of your manuscript to us. We would expect a full point-by-point response to the reviewer concerns and comments. Furthermore, Rev1, minor points 1, 3, 4 and Rev#3, criticisms pertaining to lack of quantifications should get addressed.

Given the current pandemic and lockdown situation, we understand that even such a minor revision may take some time. This is not a problem at all. Please get in touch in case you would like to discuss individual revision points further with me.

Thank you for this interesting contribution to Life Science Alliance. We are looking forward to receiving your revised manuscript.

Sincerely,

Andrea Leibfried, PhD
Executive Editor
Life Science Alliance

Meyerhofstr. 1
69117 Heidelberg, Germany
t +49 6221 8891 502
e a.leibfried@life-science-alliance.org
www.life-science-alliance.org

B. MANUSCRIPT ORGANIZATION AND FORMATTING:

We would first like to thank the reviewers for their insightful comments. We have revised the manuscript to address the raised issues. Below, find a point-by-point answer to the referees 1 and 3's comments.

Reviewer #1 (Comments to the Authors (Required)):

This manuscript describes analysis of the two different COPI gamma-COP subunit isoforms during neuronal differentiation. While it has been known for many years that two gamma-COP isoforms exist (gamma-1 and gamma-2 COP), the functional significance has remained mysterious. Here, using elegant genome editing approaches, it is shown that gamma-1 and gamma-2 COP are both required for optimum cell growth and Golgi structure, but cells remain viable upon loss of either. Analysis of neurogenesis in vitro shows that gamma-1 is more abundantly expressed than gamma-2 COP, and its loss disrupts formation of intermediate structures seen in the neurogenesis programme called embryoid bodies (EB), as well as neurite outgrowth that occurs at a later stage. Gamma-2 COP when more abundantly expressed can compensate the EB phenotype, but not neurite outgrowth, indicating a unique, paralog-specific role for gamma-1 COP in this process.

I found the results to be well presented, and convincing. The authors' conclusions were generally well supported by the data, and I found the study to be interesting. However, there are a couple of major points that lessen my enthusiasm for the work. The first is that the data are all generated using an in vitro system for neuronal differentiation. The study would be much stronger in my view if there was a demonstrable effect of gamma-1 COP manipulation in an in vivo model, where neurogenesis is occurring naturally. The second is that while the results clearly show gamma-1 COP is important for neurite outgrowth, albeit in vitro, there is a lack of mechanism to explain the phenotype (the same applies to the EB data). One might expect that disrupted trafficking is responsible, but this is not explored.

We thank the referee for finding our results convincing, our conclusions well supported by the data and our study interesting.

We agree that an in vivo model would be tremendous, unfortunately the most simple model suitable for the study of gamma-COP paralogs would be a conditional KO mice (the paralogs are only expressed in mammals and since we see a defect in EB formation it is unlikely we would get animals with a constitutive KO strategy). We also agree that addressing the mechanism underlying the phenotype will be an important next step but realistically this will take another PhD project and at least another 3-4 years of work before we address these two points.

There are also some more minor specific points as detailed below:

1.) The EM analysis of Golgi morphology in Fig 3 could be more extensive. For example, are the number of cisternae per stack or cisternal length altered by loss of gamma-1 or gamma-2 COP? Counting Golgi stacks per cell seems very subjective, especially when it seems the Golgi area of the cell is pre-selected for counting. It seems odd that the numbers are so consistent when sectioning is likely to give a high degree of variability in terms of cutting through the Golgi stacks.

Thank you for this comment. We have added a quantification of the average cisternal length in the three cell lines. For the counting of the Golgi stacks per cell, the original method section was probably misleading and was re-phrased as follow: random pictures of "30 cell profiles containing at least a Golgi stack were collected by quantification". So it is true that profiles for which a Golgi stack can be identified are

pre-selected to allow the subsequent analyses, and hence the number of counted stacks is not the true “number of stacks per cell” but still counting how many stacks are found in these pre-selected profiles is a useful parameter that gives a proxy of how many stacks are present in the cell lines. So that there is no confusion this parameter is now termed: “Golgi stacks per analyzed cell profile” (Fig. 3B). As for the consistency in the current measurements: we only included cell profiles with a visible nucleus. So, we always measured in the same area (that is, not including parts of Golgi present in the tips of the cells) and that helps to minimize size variations.

2.) The authors use BIP protein levels as a readout for ER stress (Fig 4A). This is quite an insensitive measure. Other analysis should be done to rule out an ER stress phenotype e.g. Xbp1 splicing.

3.) It is claimed that more gamma-2 COP KO cells show apoptosis than gamma-1 COP (Fig 4B). The microscopy images do not clearly show this effect, which should also be quantified.

We thank the reviewer for this comment. We also found the effect was not strong but seemed real on the microscope and though not critical for this study felt we had to report accordingly. We have now performed a real quantification assay for Caspase-3 activity. We chose to use dual color flow cytometry because it is much more quantitative and reliable than microscopy. This analysis rules out increased apoptosis in both KO cell lines compared to WT. Figure 4C and the main text have been updated accordingly.

4.) What does looser EB formation indicate? This phenotype is scored in several experiments but its significance is not obvious.

Thank you for this comment. We have added two comments on the formation of EBs on pages 10 and 11: first, on its importance during P19 cells differentiation (cell-cell contact are necessary for retinoic acid-mediated neuron formation) and second on a potential explanation for the formation of loose EBs (it might be due to a less efficient trafficking of cell adhesion proteins to the cell surface).

Reviewer #2 (Comments to the Authors (Required)):

COPI coated vesicles are critical for traffic between Golgi cisternae, a function that is required in all cells. This manuscript investigates the possibility that the paralogous gamma1 and gamma2 subunits of the COPI coat may have distinct functions through a rigorous set of genetic perturbations and rescue experiments in mouse P19 carcinoma cells as well as in neuron-like cells derived from them. Through such efforts, the authors show that although either gamma-subunit can support viability, gamma1 is selectively required to support neurite outgrowth during differentiation of the neuron like cells. Although the authors present data that makes a compelling argument in favor of a distinct function for gamma1, insight into how gamma1 is able to support neurite outgrowth is lacking.

Some concern also arises about the generalizability of key observations outside of the specific model system examined. Although the differentiation model to convert P19 carcinoma cells into neuron-like cells yields cells with many properties of neurons, it is not clear that the observations from this one model system can be generalized to make broad statements about the requirement for gamma1 in neuronal development. Thus, although the manuscript shows a neurite outgrowth

defect in the absence of gamma1, it remains unclear to what extent observations from this one neuron-like model system can be generalized to neurons in vivo. Nonetheless, the authors frequently make statements that indicate that their discoveries will apply much more generally.

Reviewer #3 (Comments to the Authors (Required)):

Comments to authors:

Overall, this is a strong manuscript showing that COPI1 is required for differentiation and neurite outgrowth in P19 cells. However, the manuscript lacks quantitative rigor, and this must be addressed before I can recommend publication.

Figure 1b) Why are only some of the COPI coatamer subunits measured in the cell lysates? Later figures show that the lab has a clean antibody to COPE, and commercial antibodies are available for both B and B'. The levels of all COPI coatamer members should be documented during differentiation.

We thank the reviewer for this comment. We have not scored all COPI subunits during differentiation for pragmatic reasons: we would have had to follow 9 different COP proteins (seven subunits including two that exist as two paralogs) + Oct-4 and BIII-tubulin as differentiation markers and GAPDH as the housekeeping marker. That makes a total of 12 proteins with the additional complication that gamma1-, gamma2-, beta- and beta'- are all around the same size (ca 100 kDa), and epsilon-COP and GAPDH are both about 35 kDa, and many available antibodies are from the same species.

This, plus the fact that the number of differentiated cells is limiting, led us to be pragmatic and look at less proteins. However, we tried to do it in a rational way. The COPI coatamer complex is a seven-subunit complex that can be subdivided in three building blocks that do not exist individually under physiological conditions but can be separated under high salt conditions (Loewe & Kreis 1995) or after chemically-induced protein modification (Pavel et al. 1998). The three building blocks are the alpha/beta'/epsilon complex, the beta/delta complex and the gamma/zeta complex. When choosing the subunits to analyze we picked subunits that belong to all three coatamer building blocks: gamma-COP for obvious reasons, as well as delta, and alpha. The choice of these three COPI subunits also allowed, considering their sizes, to probe Oct-4, BIII-tubulin and GAPDH. We had not explained this in the original manuscript, we have now added explanation in the revised version (see page 5).

If Western blot analysis is going to be used to measure increase or decrease in proteins of interest, they blots should be analyzed by densitometry (compared to a suitable housekeeping protein such as GAPDH) and the levels of the proteins from multiple gels reported with appropriate statistical methods applied. This criticism applies to all blots presented in this paper.

We thank the reviewer for this comment. We have added densitometry quantification whenever we make a statement about increasing/decreasing levels of protein.

Figure 2b) this blot shows that they are able to detect COPE by Western Blot, therefore COPE should be analyzed in figure 1 as well.

As stated above COPE and GAPDH have the same size. In 2b, we took α -tubulin as a housekeeping marker and therefore also showed COPE. In 1, we had to take

GAPDH as a housekeeping marker because we also needed to show β III-tubulin (same size as α -tubulin) as a differentiation marker, therefore we chose not to show COPE.

In addition, one has to keep in mind that what is really at stake here is the proportion of COPG1 over COPG2. Indeed, the COPI coat complex coatomer is an equimolar heptameric complex and in cells each of its seven subunits is only found as part of the complex, not as individual monomer or part of a subcomplex. So even if say COPE would be overexpressed compared to the other COP subunits at any point of the differentiation, the excess COPE would not be incorporated in coatomer and be rapidly degraded (this is also what we see when we massively overexpress COPG1 in COPG1 KO cells: this leads to the disappearance of COPG2).

Figure 4a) Again, quantification of the Western blot is required. By eye, this particular blot appears to show increased GRP-78 in the COPG2^{-/-} lines compared to P19WT controls. The more appropriate experiment might be to treat both WT and COPG1^{-/-} and COPG2^{-/-} cultures with Tunicamycin to evaluate the percent increase in Grp-78 as its unclear here whether the mutant lines are ABLE to respond to Tunicamycin at all.

We thank the reviewer for this comment. We have performed the required experiment and provide a quantification of the corresponding blots. We also added another ER stress marker (CHOP) to have another line of evidence. We do see that Copg1 KO cells react somewhat differently to ER stress than WT and Copg2 cells (less GRP78 observed but more CHOP) and this is now commented in the main text (page 8). But the original point stands that both KO cell line do not show evidence of ER stress.

Figure 4b) These results must be quantified. The authors state that "only occasional" caspase 3 staining was detected in WT cultures and "slightly more" was detected in COPg2^{-/-}. This vague, qualitative language is insufficient. Please provide concrete numbers from repeated experiments with appropriate measures of statistical significance.

We thank the reviewer for this comment. As stated above (see Reviewer 1, point 3), we now provide a flow cytometry assay which is much more reliable and quantitative for the purpose of measuring Caspase activity (New Figure 4C). With this assay we could not find evidence of increased apoptosis in the KO cell lines. The main text has been updated accordingly.

Figure 5) I would suggest including the results presented in 7a as figure 5e - as a reader, I was confused as to why this experiment wasn't performed and by the time it was presented, it felt out of place.

We thank the reviewer for this comment. The requested data were moved to Figure 5 as suggested. We took them from Supp. Fig. 3C rather than Fig. 5E (same experiments), otherwise it would have been difficult to introduce the Copg2 KI cell line at the right place.

Figure 6) the number of neurites per cell should be quantified as well as the average neurite length. The authors mention neurite number when discussing figure 8, but never present quantitation.

Thank you for this comment, we have added an estimation of the neurites per cell for Figure 6 (in new Figure S6). In this analysis we do not see a significant difference between the cell lines and have updated the main text accordingly. We report the data with a note of caution (page 14) on this parameter as P19 cells grow at relatively high densities, which can make the assignment of unique neurite attachment points (the proxy for the number of neurites) challenging.

6c/d should be quantified by densitometry to verify statement of increased protein levels.

We now provide a quantification of blots corresponding to Fig. 6C (shown on Supp. Fig. S8) to support the statement of page 15 on the increase of expression of gamma2-COP during differentiation in Copg1 KO cells compared to WT cells.

The COPG2^{-/-} cultures appear to adopt B-III tubulin positive cells earlier than P19wt cultures, but this is not addressed. Is this finding consistent? Paired with the finding in 6b that the copg2^{-/-} cultures appear to have a trend towards longer neurites, this needs to be discussed.

Thanks for pointing to this. This is not a consistent finding (compare for example, the WT samples in Fig. 6c upper and lower experiment), the signals for bIII-tubulin tend to vary from experiment to experiment but the trend is consistently the same (upregulation as differentiation is proceeding).

When discussing the results of figure 6, the authors state that in CP{G^{-/-} cells a substantial portion of the cellular coatmer simply lacks COPG. This should be addressed using the B'-COP co-IP reported in figure 2C using lysate from differentiated P19 cultures.

Thank you for this comment. We actually did not make any claim when discussing Fig. 6 but only raise the possibility that in Copg1 KO cells a substantial portion of coatmer may lack Copg. We mention that we can not exclude this possibility and we should have been more specific on why it is not so trivial to address. In the case of the b-COP depleted cells that we refer to, the experiment suggested by the referee was performed in the cited paper: IP of coatmer with the very same b'-COP antibody that we use in 2c followed by WB showing lack of b-COP signal. This is indeed neat and convincing. However, b-COP does not come as two paralogs so it is either there or not. In our case, when we deplete COPG1 we observe an upregulation of COPG2 and the question to address is if this upregulation yields the same amount of COPG as in WT cells. We do not have an antibody that recognizes COPG1 and COPG2 equally, and we are also not aware of any antibody with such a property. Hence we can not reliably estimate how much COPG2 is expressed in COPG1 KO cells in relation to COPG1 in WT cells. For this reason, we think it is more reasonable to keep the statement about coatmer lacking COPG as a possibility that we can not exclude and rephrased the main text to make it clear that it is not a claim and why it is difficult to address (page 15).

In any case, right or wrong, this hypothesis does not change the outcome of the study and was mentioned as one of our lines of thought that motivated us to perform the knock-in experiments.

Figure 8) The text declares "more and longer neurites" but only neurite length is quantified and presented. Please quantify and present neurite number with appropriate statistical methods to determine significance. This criticism applies to figure 6 as well.

Thank you for this comment. We have added an estimation of the neurites per cell (new Figure S6) and updated the main text accordingly as for Fig. 6 above.

July 3, 2020

RE: Life Science Alliance Manuscript #LSA-2020-00714-TR

Dr. Julien L Béthune
Heidelberg University
Heidelberg University Biochemistry Center (BZH)
Im Neuenheimer Feld 328
Heidelberg, Baden-Wuerttemberg 69120
Germany

Dear Dr. Béthune,

Thank you for submitting your revised manuscript entitled "A paralog-specific role of COPI vesicles in the neuronal differentiation of mouse pluripotent cells". We would be happy to publish your paper in Life Science Alliance pending final revisions necessary to meet our formatting guidelines.

- please provide supplementary figures as single files and have the supplementary figure legends as only part of the main manuscript rather than underneath the supplementary figures
- Please add scale bars to Figure 1A, Figure 5 and Figure 7
- please make scale bars in Figure 6A and Figure 8A more visible
- Figure 3A: please make sure that the zoomed in part matches the cut-out
- please check your Figure callouts (you have a callout for figure S5A, but there is not a figure s5A)
- please add a callout for Figure 3E
- Please upload your source data as a separate file and label it as source data

A. FINAL FILES:

B. MANUSCRIPT ORGANIZATION AND FORMATTING:

Sincerely,

Reilly Lorenz
Editorial Office Life Science Alliance
Meyerhofstr. 1
69117 Heidelberg, Germany
t +49 6221 8891 414
e contact@life-science-alliance.org
www.life-science-alliance.org

July 6, 2020

RE: Life Science Alliance Manuscript #LSA-2020-00714-TRR

Dr. Julien L Béthune
Heidelberg University
Heidelberg University Biochemistry Center (BZH)
Im Neuenheimer Feld 328
Heidelberg, Baden-Wuerttemberg 69120
Germany

Dear Dr. Béthune,

Thank you for submitting your Research Article entitled "A paralog-specific role of COPI vesicles in the neuronal differentiation of mouse pluripotent cells". It is a pleasure to let you know that your manuscript is now accepted for publication in Life Science Alliance. Congratulations on this interesting work.

DISTRIBUTION OF MATERIALS:

Again, congratulations on a very nice paper. I hope you found the review process to be constructive and are pleased with how the manuscript was handled editorially. We look forward to future exciting submissions from your lab.

Sincerely,

Reilly Lorenz
Editorial Office Life Science Alliance
Meyerhofstr. 1
69117 Heidelberg, Germany
t +49 6221 8891 414
e contact@life-science-alliance.org
www.life-science-alliance.org